# Characterizing the Optimal $0 - 1$ Loss for Multi-class Classification with a Test-time Attacker

**Sihui Dai**[1]*    **Wenxin Ding**[2]*    **Arjun Nitin Bhagoji**[2]    **Daniel Cullina**[3]
**Ben Y. Zhao**[2]    **Haitao Zheng**[2]    **Prateek Mittal**[1]
[1]Princeton University    [2]University of Chicago    [3]Pennsylvania State University
{sihuid,pmittal}@princeton.edu
{wenxind, abhagoji}@uchicago.edu
{ravenben, htzheng}@cs.uchicago.edu
cullina@psu.edu

## Abstract

Finding classifiers robust to adversarial examples is critical for their safe deployment. Determining the robustness of the best possible classifier under a given threat model for a fixed data distribution and comparing it to that achieved by state-of-the-art training methods is thus an important diagnostic tool. In this paper, we find achievable information-theoretic lower bounds on robust loss in the presence of a test-time attacker for *multi-class classifiers on any discrete dataset*. We provide a general framework for finding the optimal $0 - 1$ loss that revolves around the construction of a conflict hypergraph from the data and adversarial constraints. The prohibitive cost of this formulation in practice leads us to formulate other variants of the attacker-classifier game that more efficiently determine the range of the optimal loss. Our valuation shows, for the first time, an analysis of the gap to optimal robustness for classifiers in the multi-class setting on benchmark datasets.

## 1  Introduction

Developing a theoretical understanding of the vulnerability of classifiers to adversarial examples (27; 12; 8; 4) generated by a test-time attacker is critical to their safe deployment. Past work has largely taken one of two approaches. The first has focused on generalization bounds on learning in the presence of adversarial examples, by trying to determine the sample complexity of robust learning (10; 25; 21). The second has been to characterize the lowest possible loss achievable within a specific hypothesis class (22; 5; 6) for binary classification for a specified data distribution and attacker. The hypothesis class of choice is often the set of all possible classification functions. The optimal loss thus determined is a lower bound on robustness for classifiers used in practice, allowing practitioners to measure progress for defenses and step away from the attack-defense arms race (19; 31; 9).

In this paper, we take on the second approach and propose methods to find the lowest possible loss attainable by any measurable classifier in the presence of a test-time attacker in the multi-class setting. The loss thus obtained is referred to as the *optimal loss* and the corresponding classifier, the *optimal classifier*. We extend previous work (5; 6) which was restricted to the binary setting, allowing us to compare the robustness of multi-class classifiers used in practice to the optimal loss. Our **first contribution** is thus *extending the conflict-graph framework for computing lower bounds on robustness to the multi-class setting.* In this framework, given a dataset and attacker, we construct a *conflict hypergraph* which contains vertices representing training examples in the dataset, and hyperedges representing overlaps between adversarial neighborhoods around each training example. Using this hypergraph, we construct a linear program whose optimal value is a lower bound on the $0 - 1$ loss for all classifiers and whose solution is the optimal classifier. The lower bound on robustness is thus achievable.

37th Conference on Neural Information Processing Systems (NeurIPS 2023).

In practice, however, we find that the full multi-class formulation of the lower bound, although exact, can lead to prohibitively large optimization problems. Thus, we vary the information available to either the attacker or classifier to find alternative lower bounds that are quicker to compute. Our **second contribution** is the *development and analysis of more efficient methods to determine the range of loss obtained by the optimal classifier.* (see Table 1). We also interpret these methods as classifier-attacker games. In particular, we find lower bounds on the optimal loss by aggregating the set of all binary classifier-only lower bounds as well as by using truncated hypergraphs (hypergraphs with a restriction on the maximum hyperedge degree). We also upper bound the optimal loss with a generalization of the Caro-Wei bound (1) on a graph's independent set size. The gap between the lower and upper bounds on the optimal loss allows us to determine the range within which the optimal loss lies.

To analyze the performance of classifiers obtained from adversarial training (19; 31), we compare the loss obtained through adversarial training to that of the optimal classifier. We find a loss differential that is greatly exacerbated compared to the binary case (5; 22). In addition, we also determine the cases where, in practice, the bounds obtained from game variants are close to the optimal. We find that while the aggregation of binary classifier-only bounds leads to a very loose lower bound, the use of truncated hypergraphs can significantly speed up computation while achieving a loss value close to optimal. This is validated using the Caro-Wei upper bound, with the lower and upper bounds on the optimal loss closely matching for adversarial budgets used in practice. Thus, our **final contribution** is an *extensive empirical analysis of the behavior of the optimal loss for a given attacker, along with its lower and upper bounds.* This enables practitioners to utilize our methods even when the optimal loss is computationally challenging to determine.

The rest of the paper is organized as follows: §2 provides the characterization of the optimal loss; §3 proposes several upper and lower bounds on the optimal loss; §4 computes and compares the optimal loss to these bounds, as well as the performance of robustly trained classifiers and §5 concludes with a discussion of limitations and future work.

# 2   Characterizing Optimal 0-1 Loss

In this section, we characterize the optimal $0 - 1$ loss for any discrete distribution (*e.g.* training set) in the presence of a test-time attacker. This loss can be computed as the solution of a linear program (LP), which is defined based on a hypergraph constructed from the classification problem and attacker constraint specification. The solution to the LP can be used to construct a classifier achieving the optimal loss, and *lower bounds* the loss attainable by any particular learned classifier.

## 2.1   Problem Formulation

**Notation.** We consider a classification problem where inputs are sampled from input space $\mathcal{X}$, and labels belong to $K$ classes: $y \in \mathcal{Y} = [K] = \{1, ..., K\}$. Let $P$ be the joint probability over $\mathcal{X} \times \mathcal{Y}$. Let $\mathcal{H}_{\text{soft}}$ denote the space of all soft classifiers; i.e. specifically, for all $h \in \mathcal{H}_{\text{soft}}$ we have that $h : \mathcal{X} \to [0, 1]^K$ and $\sum_{i=1}^{K} h(x)_i = 1$ for all $x \in \mathcal{X}$. Here, $h(x)_i$ represents the probability that the input $x$ belongs to the $i^{\text{th}}$ class. We use the natural extension of $0 - 1$ loss to soft classifiers as our loss function: $\ell(h, (x, y)) = 1 - h(x)_y$. This reduces to $0 - 1$ loss when $h(x) \in \{0, 1\}^K$. [1]

**The adversarial classification game.** We are interested in the setting where there exists a test-time attacker that can modify any data point $x$ to generate an adversarial example $\tilde{x}$ from the neighborhood $N(x)$ of $x$. An instance of the game is specified by a discrete probability distribution $P$,[2] hypothesis class $\mathcal{H} \subseteq \mathcal{H}_{\text{soft}}$, and neighborhood function $N$. We require that for all $x \in \mathcal{X}$, $N(x)$ always contains $x$. The goal of the classifier player is to minimize expected classification loss and the goal of the

---

[1] A soft classifier can be interpreted as a randomized hard-decision classifier $f$ with $\Pr[f(x) = y] = h(\tilde{x})_y$, in which case $\ell(h, (x, y)) = \Pr[f(x) \neq y]$, classification error probability of this randomized classifier.

[2] Properly extending this game to more general data distributions is tricky due to issues with the measurability of the supremum of the loss. See Pydi and Jog for discussion of the multiple approaches taken in the literature and when various versions of the game are equivalent (23). These technicalities do not arise for discrete distributions, which are sufficient for our purposes.

attacker is to maximize it. The optimal loss is

$$L^*(P, N, \mathcal{H}) = \inf_{h \in \mathcal{H}} \mathbb{E}_{(x,y) \sim P} \Big[ \sup_{\tilde{x} \in N(x)} \ell(h, (\tilde{x}, y)) \Big] = 1 - \sup_{h \in \mathcal{H}} \mathbb{E}_{(x,y) \sim P} \Big[ \inf_{\tilde{x} \in N(x)} h(\tilde{x})_y \Big]. \quad (1)$$

**Alternative hypothesis classes.** In general, for $\mathcal{H}' \subseteq \mathcal{H}$, we have $L^*(P, N, \mathcal{H}) \leq L^*(P, N, \mathcal{H}')$. Two particular cases of this are relevant. First, the class of hard-decision classifiers is a subset of the class of soft classifiers ($\mathcal{H}_{\text{soft}}$). Second, for any fixed model parameterization (ie. fixed NN architecture), the class of functions represented by that parameterization is another subset. Thus, optimal loss over $\mathcal{H}_{\text{soft}}$ provides a lower bound on loss for these settings.

## 2.2 Optimal loss for distributions with finite support

Since we would like to compute the optimal loss for distributions $P$ *with finite support*, we can rewrite the expectation in Equation 1 as an inner product. Let $V$ be the support of $P$, i.e. the set of points $(x, y) \in \mathcal{X} \times \mathcal{Y}$ that have positive probability in $P$. Let $p \in [0, 1]^V$ be the probability mass vector for $P$: $p_v = P(\{v\})$. For a soft classifier $h$, let $q_N(h) \in \mathbb{R}^V$ be the vector of robustly correct classification probabilities for vertices $v = (x, y) \in V$, *i.e.* $q_N(h)_v := \inf_{\tilde{x} \in N(x)} h(\tilde{x})_y$. Rewriting (1) with our new notation, we have $1 - L^*(P, \mathcal{H}_{\text{soft}}, N) = \sup_{h \in \mathcal{H}_{\text{soft}}} p^T q_N(h)$. This is the maximization of a linear function over all possible vectors $q_N(h)$. In fact, the convex hull of all correct classification probability vectors is a polytope and this optimization problem is a linear program, as described next.

**Definition 1.** *For a soft classifier $h$, the correct-classification probability achieved on an example $v = (x, y)$ in the presence of an adversary with constraint $N$ is $q_N(h)_v = \inf_{\tilde{x} \in N(x)} h(\tilde{x})_y$.*

*The space of achievable correct classification probabilities is $\mathcal{P}_{\mathcal{V}, N, \mathcal{H}} \subseteq [0, 1]^{\mathcal{V}}$, defined as*

$$\mathcal{P}_{\mathcal{V}, N, \mathcal{H}} = \bigcup_{h \in \mathcal{H}} \prod_{v \in \mathcal{V}} [0, q_N(h)_v]$$

In other words we say that $q' \in [0, 1]^{\mathcal{V}}$ is achievable when there exists $h \in \mathcal{H}$ such that $q' \leq q_N(h)$. The inequality appears because we will always take nonnegative linear combinations of correct classification probabilities.

Characterizing $\mathcal{P}_{\mathcal{V}, N, \mathcal{H}}$ allows the minimum adversarial loss achievable to be expressed as an optimization problem with a linear objective:[3]

$$1 - L^*(P, N, \mathcal{H}_{\text{soft}}) = \sup_{h \in \mathcal{H}_{\text{soft}}} \mathbb{E}_{v \sim P}[q_N(h)_v] = \sup_{h \in \mathcal{H}_{\text{soft}}} p^T q_N(h) = \sup_{q \in \mathcal{P}_{\mathcal{V}, N, \mathcal{H}}} p^T q. \quad (2)$$

(6) characterized $\mathcal{P}_{\mathcal{V}, N, \mathcal{H}_{soft}}$ in the two-class case and demonstrated that this space can be captured by linear inequalities. We now demonstrate that this also holds for the multi-class setting.

## 2.3 Linear Program to obtain Optimal Loss

In order to characterize $\mathcal{P}_{\mathcal{V}, N, \mathcal{H}_{soft}}$, we represent the structure of the classification problem with a *conflict hypergraph* $\mathcal{G}_{\mathcal{V}, N} = (\mathcal{V}, \mathcal{E})$, which records intersections between neighborhoods of points in $\mathcal{X}$. The set of vertices $V$ of $\mathcal{G}_{\mathcal{V}, N}$ is the support of $P$. $\mathcal{E}$ denotes the set of hyperedges of the graph. For a set $S \subseteq \mathcal{V}$, $S \in \mathcal{E}$ (i.e. $S$ is a hyperedge in $\mathcal{G}_{\mathcal{V}, N}$) if all vertices in $S$ belong to different classes and the neighborhoods of all vertices in $S$ overlap: $\bigcap_{(x,y) \in S} N(x) \neq \emptyset$. Thus, the size of each hyperedge is at most $K$, $\mathcal{E}$ is downward-closed (meaning if $e \in \mathcal{E}$ and $e' \subset e$, then $e' \subset \mathcal{E}$), and every $v \in \mathcal{V}$ is a degree 1 hyperedge.

Using the conflict hypergraph $\mathcal{G}_{\mathcal{V}, N}$, we can now describe $\mathcal{P}_{\mathcal{V}, N, \mathcal{H}_{soft}}$.

**Theorem 1** (Feasible output probabilities (Adapted from (6))). *The set of correct classification probability vectors for support points $\mathcal{V}$, adversarial constraint $N$, and hypothesis class $\mathcal{H}_{soft}$ is*

$$\mathcal{P}_{\mathcal{V}, N, \mathcal{H}_{soft}} = \{q \in \mathbb{R}^{\mathcal{V}} : q \geq \mathbf{0}, \, Bq \leq \mathbf{1}\} \quad (3)$$

*where $B \in \mathbb{R}^{\mathcal{E} \times \mathcal{V}}$ is the hyperedge incidence matrix of the conflict hypergraph $G_{\mathcal{V}, N}$.*

---

[3]For any loss function that is a decreasing function of $h(\tilde{x})_y$, the optimal loss can be specified as an optimization over $\mathcal{P}_{\mathcal{V}, N, \mathcal{H}}$. In fact (6) focused on cross-entropy loss, which has this property.

| | Summary of Method | Location |
|---|---|---|
| **Optimal 0-1 loss** | LP on conflict hypergraph | §2.3 |
| **Lower bounds for optimal 0-1 loss** | LP on truncated conflict hypergraph | §3.1 |
| | Combining binary classification bounds | §3.2 |
| **Upper bound for optimal 0-1 loss** | Generalization of Caro-Wei bound | §3.5 |

Table 1: Summary of methods for computing the optimal $0-1$ loss and efficient bounds on this value.

See Supplementary §A for proof. This characterization of $\mathcal{P}_{\mathcal{V},N,\mathcal{H}_{soft}}$ allows us to express optimal loss as a linear program for any dataset and attacker using Eq. 2 [4].

**Corollary 1** (Optimal loss as an LP). *For any distribution $P$ with finite support,*

$$1 - L^*(P, N, \mathcal{H}_{soft}) = \max_q p^T q \ \ s.t \ q \geq 0, \ Bq \leq 1. \tag{4}$$

**Duality and adversarial strategies.** The dual linear program is

$$\min_z \mathbf{1}^T z \quad \text{s.t.} \quad z \geq 0, \quad B^T z \geq p.$$

Feasible points in the dual linear program are fractional coverings of the vertices by the hyperedges (24). (See Section 3.4 and Supplementary §B for more discussion of fractional packings and coverings.) An adversarial strategy can be constructed from a feasible point $z$ as follows. For each hyperedge $e$, chose an example $\tilde{x}(e)$ such that $\tilde{x}(e) \in N(x)$ for all $(x, y) \in e$. From the definition of the conflict hypergraph, a choice is always available. A randomized adversarial strategy consists of conditional distributions for the adversarial example $\tilde{x}$ given the natural example $x$. When the adversary samples the natural example $(x, y)$, the adversary can select $\tilde{x}(e)$ with probability at least $\frac{B_{e,v} z_e}{p_v}$. Note that $B_{e,v} z_e$ is the amount of coverage of $v$ coming from $e$. This is nonzero only for $e$ that contain $(x, y)$. A conditional distribution satisfying these inequalities exists because $\sum_e B_{e,v} z_e = (B^T z)_v \geq p_v$.

Thus for a vertex $v$ such that $(B^T z)_v = p_v$, there is only one choice for this distribution. For a vertex $v$ that is over-covered, i.e. $(B^T z)_v > p_v$, the adversary has some flexibility. If $v$ is over-covered in some minimal cost covering, by complementary slackness $q_v = 0$ in every optimal $q$, so the optimal classifiers do not attempt to classify $v$ correctly.

**Three-class minimal examples.** Corollary 1 demonstrates that the optimal loss for the multi-class problem is influenced by hyperedges between vertices which reflect higher order interactions between examples. Figure 1 shows an important distinction between two types of three-way interactions of examples.

We have $1 - L^*(P, \mathcal{H}_{soft}, N) = \max(p_u, p_v, p_w, \frac{1}{2})$ while $1 - L^*(P', \mathcal{H}_{soft}, N) = \max(p'_u, p'_v, p'_w)$. The presence or absence of the size-three hyperedge affects the optimal loss if and only if the example probabilities are close to balanced, i.e. all at most $\frac{1}{2}$.

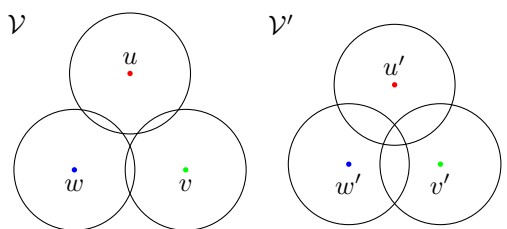

Figure 1: Two possible conflict structures involving three examples, each from a different class. In the right case, all subsets of $\mathcal{V}' = \{u', v', w'\}$ are hyperedges in the conflict hypergraph $\mathcal{G}_{\mathcal{V}',N}$. In the left case, $\{u, v, w\}$ is not a hyperedge in $\mathcal{G}_{\mathcal{V},N}$, but all other subsets of $\mathcal{V}$ are.

It is instructive to consider the optimal classifiers and adversarial strategies in the two cases. For $\mathcal{V}$, when $\frac{1}{2} \leq p_u$, the classifier $h$ with $q_N(h) = (1, 0, 0)$ is optimal. One such classifier is the constant classifier $h(x) = (1, 0, 0)$. The optimal cover satisfies $z_{\{u,v\}} + z_{\{u,w\}} = p_u, z_{\{u,v\}} \geq p_v, z_{\{u,w\}} \geq p_w, z_{\{v,w\}} = 0$. Thus when the adversary

---
[4]We note that concurrent work by Trillos et al. (29) provides a general-purpose framework based on optimal transport to find lower bounds on robustness in the per-sample as well as distributional sense. They also independently construct the three-class minimal example from Figure 1.

samples $v$ or $w$, it always produces an adversarial example that could be confused for $u$. When $\max(p_u, p_v, p_w) \leq \frac{1}{2}$, any classifier $h$ with $q_N(h) = (\frac{1}{2}, \frac{1}{2}, \frac{1}{2})$ is optimal. To achieve these correct classification probabilities, we need $h(\tilde{x}) = (\frac{1}{2}, \frac{1}{2}, 0)$ for $\tilde{x} \in N(u) \cap N(v)$, $h(\tilde{x}) = (\frac{1}{2}, 0, \frac{1}{2})$ for $\tilde{x} \in N(u) \cap N(w)$, etc.. The cover $z_{\{u,v\}} = p_u + p_v - \frac{1}{2}$, $z_{\{u,w\}} = p_u + p_w - \frac{1}{2}$, and $z_{\{v,w\}} = p_v + p_w - \frac{1}{2}$ is optimal and has cost $\frac{1}{2}$. The adversary produces examples associated with all three edges.

For $\mathcal{V}'$, things are simpler. The cover $z_{\{u,v,w\}} = \max(p_u, p_v, p_w)$ is always optimal. When $p_u \geq \max(p_v, p_w)$, the classifier that returns $(1, 0, 0)$ everywhere is optimal.

# 3 Bounding the Optimal 0-1 Loss

While Corollary 1 characterizes the optimal loss, it may be computationally expensive to construct conflict hypergraphs in practice for a given dataset and to solve the linear program. Thus, we discuss several methods of bounding the optimal loss from the LP in Corollary 1, which are computationally faster in practice (§4.2).

## 3.1 Lower bounds on multiclass optimal loss via truncated hypergraphs

The edge set of the hypergraph $\mathcal{G}$ can be very large: there are $\prod_{i \in [K]}(1 + |V_i|)$ vertex sets that are potential hyperedges. Even when the size of the edge set is reasonable, it is not clear that higher order hyperedges can be computed from $\mathcal{V}$ efficiently. To work around these issues, we consider hypergraphs with bounded size hyperedges: $\mathcal{G}^{\leq m} = (\mathcal{V}, \mathcal{E}^{\leq m})$ where $\mathcal{E}^{\leq m} = \{e \in \mathcal{E} : |e| \leq m\}$. We refer to these hypergraphs as *truncated hypergraphs*. In the corresponding relaxation of (4), $B$ is replaced by $B^{\leq m}$, the incidence matrix for $\mathcal{E}^{\leq m}$. Since $\mathcal{E}^{\leq m} \subseteq \mathcal{E}$, this relaxation provides a lower bound on $L^*(P, N, \mathcal{H}_{\text{soft}})$.

**Classification with side-information.** This relaxation has an interpretation as the optimal loss in a variation of the classification game with side information.

**Definition 2.** *In the example-dependent side information game with list length $m$, the adversary samples $(x, y) \sim P$, then selects $\tilde{x}$ and $C \subseteq \mathcal{Y}$ such that $y \in C$ and $|C| = m$. We call $C$ the side information. The classifier receives both $\tilde{x}$ and $C$, so the classifier is a function $h : \mathcal{X} \times \binom{\mathcal{Y}}{m} \to [0, 1]^K$, where $\binom{\mathcal{Y}}{m}$ is the set of $m$-element subsets of $\mathcal{Y}$. Let*

$$L^*(m, P, \mathcal{H}, N) = \inf_{h \in \mathcal{H}} \mathbb{E}_{(x,y) \sim P}\left[\inf_{\tilde{x} \in N(x)} \min_{C \in \binom{\mathcal{Y}}{m} : y \in C} (1 - h(\tilde{x}, C)_y)\right]$$

*be the minimum loss in this game.*

To illustrate this, consider the distribution $P'$ from Figure 1 with $m = 2$. The adversary can select some $\tilde{x} \in N(u') \cap N(v') \cap N(w')$, but the classifier will use the side-information to eliminate one of the three classes. The classifier is in the same situation it would be if the distribution were $P$ and the size-three hyperedge was absent.

**Definition 3.** *For classifiers using class list side-information, the correct-classification probability is defined as follows: $q_{m,N}(h)_{(x,y)} = \inf_{\tilde{x} \in N(x)} \min_{C \in \binom{[K]}{m} : y \in C} h(\tilde{x}, C)_y$. The set of achievable correct-classification probabilities is $\mathcal{P}_{m,\mathcal{V},N,\mathcal{H}} = \bigcup_{h \in \mathcal{H}} \prod_{v \in \mathcal{V}}[0, q_{m,N}(h)_v]$.*

When $m = K$, the minimization over $C$ is trivial and $q_{m,N}(h)$ reduces to $q_N(h)$.

**Theorem 2** (Feasible output probabilities in the side-information game)**.** *The set of correct classification probability vectors for side-information of size $m$, support points $\mathcal{V}$, adversarial constraint $N$, and hypothesis class $\mathcal{H}_{soft}$ is*

$$\mathcal{P}_{m,\mathcal{V},N,\mathcal{H}_{soft}} = \{q \in \mathbb{R}^{\mathcal{V}} : q \geq \mathbf{0}, B^{\leq m} q \leq \mathbf{1}\} \tag{5}$$

*where $B^{\leq m} \in \mathbb{R}^{\mathcal{E} \times \mathcal{V}}$ is the hyperedge incidence matrix of the conflict hypergraph $G_{\mathcal{V},N}^{\leq m}$.*

The proof can be found in Supplementary §A.

Using the feasible correct classification probabilities in Theorem 2, we can now write the LP for obtaining the optimal loss for classification with side-information:

**Corollary 2** (Optimal loss for classification with side information / truncation lower bound).

$$1 - L^*(P, N, \mathcal{H}_{soft}) = \max_q p^T q \ \ s.t \ q \geq 0, \ B^{\leq m} q \leq 1.$$

## 3.2 Lower bounds on multiclass optimal loss via lower bounds for binary classification

For large training datasets and large perturbation sizes, it may still be computationally expensive to compute lower bounds via LP even when using truncated hypergraphs due to the large number of edge constraints. Prior works (5; 6) proposed methods of computing lower bounds for $0 - 1$ loss for binary classification problems and demonstrate that their algorithm is more efficient than generic LP solvers. We now ask the question: *Can we use lower bounds for binary classification problems to efficiently compute a lower bound for multi-class classification?*

Consider the setting where we obtain the optimal $0-1$ loss for all one-versus-one binary classification tasks. Specifically, for each $C \in \binom{[K]}{2}$, the binary classification task for that class pair uses example distribution $P|Y \in C$ and the corresponding optimal loss is $L^*((P|Y \in C), N, \mathcal{H}_{soft})$. What can we say about $L^*(P, N, \mathcal{H}_{soft})$ given these $\binom{K}{2}$ numbers?

This question turns about to be related to another variation of classification with side information.

**Definition 4.** *In the* class-only side-information *game, the adversary samples $y \sim P_y$, then selects $C \in \binom{[K]}{m}$, then samples $x \sim P_{x|y}$ and selects $\tilde{x} \in N(x)$. Let $L^*_{co}(m, P, \mathcal{H}, N)$ be the minimum loss in this game.*

In the example-dependent side-information game from Section 3.1, the adversary's choice of $C$ can depend on both $x$ and $y$. In the class-only variation it can only depend on $y$. For the class only game, we will focus on the $m = 2$ case.

To make the connection to the binary games, we need to add one more restriction on the adversary's choice of side information: for all $y, y'$ $\Pr[C = \{y, y'\}|Y = y] = \Pr[C = \{y, y'\}|Y = y']$. This ensures that the classifier's posterior for $Y$ given $C$ is $\Pr[Y = y|C] = \Pr[Y = y]/\Pr[Y \in C]$.

**Theorem 3.** *The optimal loss in the class-only side-information game is $L^*_{co}(2, P, N, \mathcal{H}) = \max_s \sum_{i,j} Pr[Y = i] a_{i,j} s_{i,j}$ where $a_{i,j} = L^*(P|(y \in \{i,j\}), \mathcal{H}, N)$ and $s \in \mathbb{R}^{[K] \times [K]}$ is a symmetric doubly stochastic matrix: $s \geq 0$, $s = s^T$, $s\mathbf{1} = \mathbf{1}$.*

The proof in is Supplementary §A. The variable $s$ represents the attacker's strategy for selecting the class side information. When the classes are equally likely, we have a maximum weight coupling problem: because the weights $a$ are symmetric, the constraint that $s$ be symmetric becomes irrelevant.

## 3.3 Relationships between games and bounds

The side information games provide a collection of lower bounds on $L^*(P, N, \mathcal{H}_{soft})$. When $m = 1$, the side infomation game becomes trivial: $C = \{y\}$ and the side information contains the answer to the classification problem. Thus $L^*(1, \mathcal{H}_{soft}) = L^*_{co}(1, \mathcal{H}_{soft}) = 0$. When $m = K$, $C = \mathcal{Y}$ and both the example-dependent and class-only side information games are equivalent to the original game, so $L^*(P, N, \mathcal{H}) = L^*(K, P, N, \mathcal{H}) = L^*_{co}(K, P, N, \mathcal{H})$. For each variation of the side-information game, the game becomes more favorable for the adversary as $m$ increases: $L^*(m, P, n, \mathcal{H}) \leq L^*(m+1, P, N, \mathcal{H})$ and $L^*_{co}(m, P, N, \mathcal{H}) \leq L^*_{co}(m+1, P, N, \mathcal{H})$. For each $m$, it is more favorable for the adversary to see $x$ before selecting $C$, *i.e.* $L^*_{co}(m, P, N, \mathcal{H}) \leq L^*(m, P, N, \mathcal{H})$.

## 3.4 Optimal Loss for Hard Classifiers

Since $\mathcal{H}_{hard} \subset \mathcal{H}_{soft}$, $L^*(m, P, N, \mathcal{H}_{soft}) \leq L^*(P, N, \mathcal{H}_{hard})$. The optimal loss over hard classifiers is interesting both as a bound on the optimal loss over soft classifiers and as an independent quantity. Upper bounds on $L^*(P, N, \mathcal{H}_{hard})$ can be combined with lower bounds from §3.1 and §3.2 using small values of $m$ to pin down $L^*(m, P, N, \mathcal{H}_{soft})$ and establish that larger choices of $m$ would not provide much additional information.

A hard classifier $h : \mathcal{X} \to \{0, 1\}^{[K]}$ has $0, 1$-valued correct classification probabilities. When we apply the classifier construction procedure from the proof of Theorem 1 using an integer-valued

$q$ vector, we obtain a hard classifier. Thus the possible correct classification probabilities for hard classifiers are $\mathcal{P}_{\mathcal{V},N,\mathcal{H}_{soft}} \cap \{0,1\}^{[K]}$. These are exactly the indicator vectors for the independent sets in $\mathcal{G}^{\leq 2}$: the vertices included in the independent set are classified correctly and the remainder are not. Formally, we can express hard classifier loss as:

$$1 - L^*(P, N, \mathcal{H}_{\text{hard}}) = \max_{S \subseteq \mathcal{V}: S \text{ independent in } \mathcal{G}^{\leq 2}} P(S). \tag{6}$$

Finding the maximum weight independent set is an NP hard problem, which makes it computationally inefficient to compute optimal hard classifier loss.

**Two-class versus Multi-class hard classification**   There are a number of related but distinct polytopes associated with the vertices of a hypergraph (24). The distinctions between these concepts explain some key differences between two-class and multi-class adversarial classification. See Supplementary §B for full definitions of these polytopes.

When $K = 2$, the conflict hypergraph is a bipartite graph. For bipartite graphs, the fractional vertex packing polytope, which has a constraint $\sum_{i \in e} q_i \leq 1$ for each edge $e$, coincides with the independent set polytope, which is the convex hull of the independent set indicators. Due to this, in the two class setting, $\mathcal{P}_{\mathcal{V},N,\mathcal{H}_{soft}}$ is the convex hull of $\mathcal{P}_{\mathcal{V},N,\mathcal{H}_{hard}}$, hard classifiers achieve the optimal loss, and optimal hard classifiers can be found efficiently.

As seen in Theorem 1, for all $K$ the fractional vertex packing polytope characterizes performance in soft classification problem. However, for $K > 2$, it becomes distinct from the independent set polytope. An independent set in a hypergraph is a subset of vertices that induces no hyperedges. In other words, in each hyperedge of size $m$, at most $m - 1$ vertices can be included in any independent set. Because the edge set of the conflict hypergraph is downward-closed, only the size-two hyperedges provide binding constraints: the independent sets in $\mathcal{G}$ are the same as the independent sets in $\mathcal{G}^{\leq 2}$. Thus the concept of a hypergraph independent set is not truly relevant for our application.

There is a third related polytope: the fractional independent set polytope of $\mathcal{G}^{\leq 2}$, which has a constraint $\sum_{i \in S} q_i \leq 1$ for each clique $S$ in $\mathcal{G}^{\leq 2}$. The fractional independent set polytope of $\mathcal{G}^{\leq 2}$ is contained in the fractional vertex packing polytope of $\mathcal{G}$: every hyperedge in $\mathcal{G}$ produces a clique in $\mathcal{G}^{\leq 2}$ but not the reverse. This inclusion could be used to find an upper bound on optimal soft classification loss.

Furthermore, when $K > 2$ the fractional vertex packing polytope of the conflict hypergraph, i.e. $\mathcal{P}_{\mathcal{V},N,\mathcal{H}_{soft}}$, can have non-integral extreme points and thus can be strictly larger than the independent set polytope. The first configuration in Figure 1 illustrates this. Thus the soft and hard classification problems involve optimization over different spaces of correct classification probabilities. Furthermore, maximum weight or even approximately maximum weight independent sets cannot be efficiently found in general graphs: the independent set polytope is not easy to optimize over.

In Section 3.5, we will use an efficiently computable lower bound on graph independence number.

### 3.5   Upper bounds on hard classifier loss via Caro-Wei bound on independent set probability

In §3.1 and 3.2, we discussed 2 ways of obtaining lower bounds for the loss of soft classifiers for the multi-class classification problem. In this section, we provide an upper bound on the loss of the optimal hard classifier (we note that this is also an upper bound for optimal loss for soft classifiers). In §3.4, we discussed the relationship between optimal loss achievable by hard classifiers and independent set size. We upper bound the optimal loss of hard classifiers by providing a lower bound on the probability of independent set in the conflict graph.

The following theorem is a generalization of the Caro-Wei theorem (1) and gives a lower bound on the weight of the maximum weight independent set.

**Theorem 4.** *Let $\mathcal{G}$ be a graph on $\mathcal{V}$ with adjacency matrix $A \in \{0,1\}^{\mathcal{V} \times \mathcal{V}}$ and let $P$ be a probability distribution on $\mathcal{V}$. For any $w \in \mathbb{R}^{\mathcal{V}}$, $w \geq 0$, there is some independent set $S \subseteq \mathcal{V}$ with*

$$P(S) \geq \sum_{v \in \mathcal{V}: w_v > 0} \frac{p_v w_v}{((A+I)w)_v}.$$

The proof is in Supplementary §A.

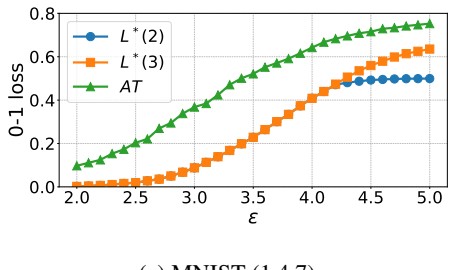
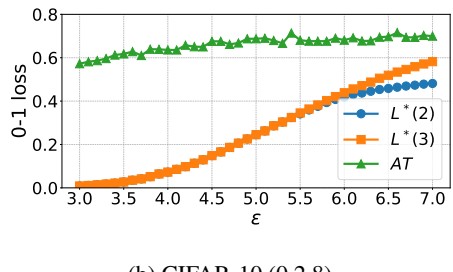

(a) MNIST (1,4,7)  (b) CIFAR-10 (0,2,8)

Figure 2: Optimal error for MNIST and CIFAR-10 3-class problems ($L^*(3)$). $L^*(2)$ is a lower bound computed using only constraints from edges. *AT* is the loss for an adversarially trained classifier under the strong APGD attack (9).

For comparison, the standard version of the Caro-Wei theorem is a simple lower bound on the independence number of a graph. It states that $\mathcal{G}$ contains an independent set $S$ with $|S| \geq \sum_{v \in \mathcal{V}} 1/(d_v + 1)$ where $d_v$ is the degree of vertex $v$.

Note that if $w$ is the indicator vector for an independent set $S'$, the bound becomes $p^T w = P(S')$. In general, the proof of Theorem 4 can be thought of as a randomized procedure for rounding an arbitrary vector into an independent set indicator vector. Vectors $w$ that are nearly independent set indicators yield better bounds.

Theorem 4 provides a lower bound on the size of the maximum independent set in $\mathcal{G}^{\leq 2}$ and thus an upper bound on $L^*(P, n, \mathcal{H}_{hard})$, which we call $L_{CW} = 1 - P(S)$.

# 4 Empirical Results

In this section, we compute optimal losses in the presence of an $\ell_2$-constrained attacker (12) at various strengths $\epsilon$ for benchmark computer vision datasets like MNIST and CIFAR-10 in a 3-class setting. We compare these optimal losses to those obtained by state-of-the-art adversarially trained classifiers, showing a large gap. We then compare the optimal loss in the 10-class setting to its upper and lower bounds (§3.3), showing matching bounds at lower $\epsilon$. In the Supplementary, §C describes hyperedge finding in practice, §D details the experimental setup and §E contains additional results.

## 4.1 Optimal loss for 3-class problems

Corollary 1 allows us to compute the optimal loss given any dataset (and its corresponding conflict hypergraph). We compute the optimal loss for 3-way classification, due to the computational complexity of finding higher order hyperedges (see §D.4 of the Supp.). In Figure 2, we plot the optimal loss $L^*(3)$ computed via the LP in Corollary 1 against the loss obtained through TRADES (31) for 3-class MNIST (classes '1', '4', '7') and 3-class CIFAR-10 ('plane', 'bird' and 'ship' classes) with 1000 samples per class [5]. For MNIST, we train a 3 layer CNN for 20 epochs with TRADES regularization strength $\beta = 1$, and for CIFAR-10, we train a WRN-28-10 model for 100 epochs with $\beta = 6$. We evaluate models using APGD-CE from AutoAttack (9).

From Figure 2, we observe that this gap is quite large even at lower values of $\epsilon$. This indicates that there is considerable progress to be made for current robust training methods, and this gap may be due to either the expressiveness of the hypothesis class or problems with optimization. We find that for CIFAR-10, TRADES is unable to achieve loss much better than 0.6 at an $\epsilon$ for which the optimal loss is near 0. This gap is much larger than observed by prior work (5; 6) for binary classification, suggesting that *current robust training techniques struggle more to fit training data with more classes.* In §D.8 of the Supp., we ablate over larger architectures, finding only small improvements at lower values of $\epsilon$ and none at higher.

---

[5]We also used PGD adversarial training (19) but found its performance to be worse (See Appendix D.6)

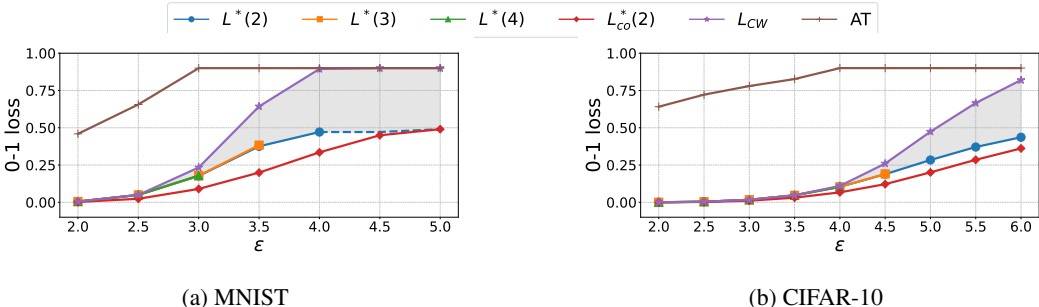

(a) MNIST                            (b) CIFAR-10

Figure 3: Lower bounds on the exact optimal 10-class loss using hyperedges up to degree 2 ($L^*(2)$), 3 ($L^*(3)$) and 4 ($L^*(4)$), as well as maximum weight coupling of pairs of binary $0 - 1$ loss lower bounds ($L^*_{co}(2)$). $L_{CW}$ is an upper bound from the Caro-Wei approximation of the independent set number. The region in grey represents the range of values we would expect the true optimal loss $L^*(10)$ to fall under.

## 4.2   Bounds on optimal loss for 10-class problems

As the number of classes and dataset size increases, the difficulty of solving the LP in Corollary 1 increases to the point of computational infeasibility (see §D.4 of the Supp.). We use the methods discussed in Section 3 to bound the optimal loss for 10-class problems on MNIST and CIFAR-10 datasets on the full training dataset. We present results for each approximation in Figure 3, with the limitation that for some methods, we are only able to obtain results for smaller values of $\epsilon$ due to runtime blowup. We provide truncated hypergraph bounds for CIFAR-100 in §D.2 of the Supp.

**Lower bounding the optimal loss using truncated hypergraphs (§3.1):** In Figure 3, we plot the loss lower bound obtained via by truncating the hypergraph to consider only edges ($L^*(2)$), up to degree 3 hyperedges ($L^*(3)$), and up to degree 4 hyperedges ($L^*(4)$). Computing the optimal loss would require computing degree 10 hyperedges. However, we find at small values of $\epsilon$, there is little difference in these bounds despite the presence of many higher degree hyperedges. This indicates that the use of use of higher degree hyperedges may not be critical to get a reasonable estimate of the optimal loss. For example, for CIFAR-10 at $\epsilon = 3$, we observe 3M degree 3 hyperedges and 10M degree 4 hyperedges, but these constraints have no impact on the computed lower bound. To understand the impact of hyperedges, we provide the count of hyperedges for each value of $\epsilon$ and plots of the distribution of optimal classification probabilities per vertex in §D.3 of the Supp. From Figure 2, we find that the difference $L^*(2)$ and $L^*(3)$ does not occur until the loss reaches above 0.4 for the 3-class problem.

*Takeaway:* In practice, we *do not lose information from computing lower bounds with only edges in the conflict hypergraph.*

**Lower bounding the optimal loss using the $1\text{v}1$ binary classification problems (§3.2):** We can use the algorithm from Bhagoji et al.(6) to efficiently compute 1v1 pairwise optimal losses to find a lower bound on the 10-class optimal loss ($L^*_{CO}(2)$). From Theorem 3, we use maximum weight coupling over these optimal losses to find a lower bound. Optimal loss heatmaps for each pair of classes in MNIST and CIFAR-10 are in §D.5 of the Supp. The efficiency of the 1v1 computation allows us to compute lower bounds at larger values of $\epsilon$ in Figure 3.

*Takeaway:* From Figure 3, we find that *while this lower bound is the most efficient to compute, the obtained bound is much looser compared to that from truncated hypergraphs.* This arises from the weak attacker assumed while computing this bound.

**Upper bounding optimal loss via Caro-Wei approximation (§3.5):** In Figure 3, we also plot the upper bound on $0 - 1$ loss for hard classifiers (denoted by $L_{CW}$) obtained via applying Theorem 4 with vertex weights obtained from the solution to $L^*(2)$. When $\epsilon$ becomes large ($\epsilon \geq 3.0$ for MNIST and $\epsilon \geq 4.5$ for CIFAR-10), the loss upper bound increases sharply. This indicates the lower bound on the independent set size becomes looser as the number of edges increases, due to the importance of higher-order interactions that are not captured by the approximation. At the small values of $\epsilon$ used in practice however, the lower bounds obtained through truncated hypergraphs ($L^*(2)$, $L^*(3)$, and $L^*(4)$) are close to the value of this upper bound .

*Takeaways:* (i) We do not lose much information from not including all hyperedges at small $\epsilon$ values as *the upper and lower bounds are almost tight*; (ii) At these small values of $\epsilon$, *we do not expect much difference in performance between hard classifiers and soft classifiers.*

**Comparing loss of trained classifiers to optimal:** In Figure 3, we also compare the loss obtained by a robustly trained classifier (AT) to our bounds on optimal loss. For both datasets, we see a large gap between the performance of adversarial training and our bounds (including the upper bound from the Caro-Wei approximation $L_{CW}$), even at small values of $\epsilon$. This suggests that current robust training techniques are currently unable to optimally fit the training data in multiclass classification tasks of interest. In addition, we also checked the performance of state-of-the-art verifiably robust models on these two datasets from a leaderboard (18). For MNIST, the best certifiably robust model has a $0 - 1$ loss of $0.27$ at a budget of $1.52$ and $0.44$ at a budget of $2.0$, while for CIFAR-10, the best certifiably robust model has a $0 - 1$ loss of $0.6$ at a budget of $1.0$ and $0.8$ at a budget of $2.0$. These are much higher than the optimal lower bound that is achievable for these datasets which is $0$ in all these cases.

*Takeaway:* Performance of state-of-the-art robust models, both empirical and verifiable, *exhibit a large gap from the range of values predicted by our bounds on optimal* $0 - 1$ *loss, even when $\epsilon$ is small.* Future research focusing on developing algorithms to decrease this gap while maintaining generalization capabilities may lead to improvements in model robustness.

## 5 Discussion and Related Work

**Related Work:** When the data distribution satisfies certain properties, Dohmatob (11) and Mahloujifar *et al.* (20) use the 'blowup' property to determine bounds on the robust loss, given some level of loss on benign data. We note that these papers use a different loss function that depends on the original classification output on benign data, thus their bounds are not comparable. Bhagoji *et al.* (5; 6), and Pydi *et al.* (22) provide lower bounds on robust loss when the set of classifiers under consideration is all measurable functions. These works *only provide bounds in the binary classification setting*. Work on verifying robustness (7; 28; 13; 18) provides bounds on the robustness of specific classifiers. Yang *et al.* (30) independently introduced the concept of a 'conflict graph' to obtain robust non-parametric classifiers via an adversarial pruning defense. The closest related work to ours is Trillos *et al.* (29), which uses optimal transport theory to find lower bounds on multi-class classification that are applicable for both continuous and discrete distributions. While their theoretical bounds are exact and more general than ours, accounting for distributional adversaries, their numerically computed bounds via the Sinkhorn algorithm are approximate and converge to the true value only as the entropy regularization decreases. In contrast, we provide methods to directly compute the optimal loss for discrete distributions, along with efficient methods for determining its range to overcome the computational bottlenecks encountered.

**Discussion and Limitations:** Our work in this paper firmly establishes for the multi-class case what was known only in the binary setting before: *there exists a large gap in the performance of current robust classifiers and the optimal classifier*. It also provides methods to bound the loss efficiently in practice, giving practitioners quick means to determine the gap. The question then arises: *why does this gap arise and how can we improve training to decrease this gap?* This paper, however, does not tackle the problem of actually closing this gap. Possible methods include increasing the architecture size (26), using additional data (15) and using soft-labels (6). A surprising finding from our experiments was that the addition of hyperedges to the multi-way conflict graph did not change the lower bounds much, indicating we are in a regime where multi-way intersections minimally impact optimal probabilities. One major limitation of our work is the computational expense at larger budgets, sample sizes and class sizes. We suspect this is due to the general-purpose nature of the solvers we use and future work should look into developing custom algorithms to speed up the determination of lower bounds.

## Acknowledgments and Disclosure of Funding

ANB, WD and BYZ were supported in part by NSF grants CNS-2241303, CNS-1949650, and the DARPA GARD program. SD was supported in part by the National Science Foundation Graduate Research Fellowship under Grant No. DGE-2039656. Any opinions, findings, and conclusions or recommendations expressed in this material are those of the author(s) and do not necessarily reflect

the views of the National Science Foundation. SD and PM were supported in part by the National Science Foundation under grant CNS-2131938, the ARL's Army Artificial Intelligence Innovation Institute (A2I2), Schmidt DataX award, and Princeton E-ffiliates Award.

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

We first present proofs for theorems from the main body in §A. We then explain fractional packing and covering in §B. We present our algorithm for hyperedge finding in §C. §D contains further details about the experimental setup with all additional experimental results in §E.

## A   Proofs

*Proof of Theorem 1.* This follows immediately from Theorem 2 with $m = K$. □

*Proof of Theorem 2.* Recall that the definition of the correct-classification probabilities is

$$q_{m,N}(h)_{(x,y)} = \inf_{\tilde{x} \in N(x)} \min_{C \in \binom{[K]}{m}: y \in C} h(\tilde{x}, C)_y$$

and the set of achievable correct-classification probabilities is

$$\mathcal{P}_{\mathcal{V},N,\mathcal{H}} = \bigcup_{h \in \mathcal{H}} \prod_{v \in \mathcal{V}} [0, q_N(h)_v].$$

First, we will show $\mathcal{P}_{m,\mathcal{V},N,\mathcal{H}_{soft}} \subseteq \{q \in \mathbb{R}^{\mathcal{V}} : q \geq \mathbf{0}, B^{\leq m} q \leq \mathbf{1}\}$. The constraint $h \in \mathcal{H}_{soft}$, $\mathbf{0} \leq q_{m,N}(h) \leq \mathbf{1}$ holds because classification probabilities $h(\tilde{x}, C)_y$ must lie in the range $[0, 1]$.

We will now demonstrate that the constraint $B^{\leq m} q \leq \mathbf{1}$ must also hold. Let $e = ((x_1, y_1), ..., (x_\ell, y_\ell))$ be a size-$\ell$ hyperedge in $\mathcal{E}^{(\leq m)}$. By construction of $\mathcal{E}^{(\leq m)}$, there exists some $\tilde{x} \in \bigcap_{i=1}^{\ell} N(x_i)$. Let $S \in \binom{[K]}{m}$ be some superset of $\{y_1, \ldots, y_\ell\}$, which exists because $\ell \leq m$. From the definition of $q_{m,N}(h)$, we have that $q_{m,N}(h)_{(x_i,y_i)} \leq h(\tilde{x}, S)_{y_i}$ for each $1 \leq i \leq \ell$. Thus,

$$\sum_{i=1}^{\ell} q_{m,N}(h)_{(x_i,y_i)} \leq \sum_{i=1}^{\ell} h(\tilde{x}, S)_{y_i} \leq \sum_{j \in \mathcal{Y}} h(\tilde{x}, S)_j = 1.$$

This gives $(B^{\leq m} q)_e \leq 1$.

Now we will show $\mathcal{P}_{m,\mathcal{V},N,\mathcal{H}_{soft}} \supseteq \{q \in \mathbb{R}^{\mathcal{V}} : q \geq \mathbf{0}, B^{\leq m} q \leq \mathbf{1}\}$.

For any vector $q$ in the polytope, we have a classifier $h : \mathcal{X} \times \binom{[K]}{m} \to \mathbb{R}^{[K]}$ that achieves at least those correct classification probabilities. This mean that $h$ has the following properties. First, $h(\tilde{x}, L)_y \geq 0$ and $\sum_{y \in [K]} h(\tilde{x}, L)_y = 1$. Second, for all $(x, y) \in \mathcal{V}$, all $\tilde{x} \in N(x)$, and all $L \in \binom{[K]}{m}$ such that $y \in L$, we have $h(\tilde{x}, L)_y \geq q_{(x,y)}$.

To get $h$, first define the function $g : \mathcal{X} \times \binom{[K]}{m} \to \mathbb{R}^{[K]}$ so $g(\tilde{x}, L)_y = 0$ for $i \notin L$ and $g(\tilde{x}, L)_y = \max(0, \sup\{q_{(x_y,y)} : x_y \in \mathcal{V}_y, \tilde{x} \in N(x_y)\})$. Let $L' \subseteq L$ be the set of indices where $g(\tilde{x}, L)_y > 0$. Then any list of vertices $e = (x_y : y \in L', x_y \in \mathcal{V}_y, \tilde{x} \in N(x_y))$ forms a hyperedge of size $|L'| \leq m$. Thus

$$\sum_{y \in [K]} g(\tilde{x}, L)_y = \sum_{y \in L'} g(\tilde{x}, L)_y = \sup_e \sum_{y \in L'} q_{(x_y,y)} \leq \sup_e 1 = 1.$$

To produce $h$, allocate the remaining probability $(1 - \sum_y g(\tilde{x}, L)_y)$ to an arbitrary class. □

*Proof of Theorem 3.* The first part of this proof applies for any side-information size $m$. The adversarial strategy for selecting $C$ is a specified by a conditional p.m.f. $p_{C|y}(C|y)$. Thus $p_{y|C}(y|C) = p_{C|y}(C|y)p_{\mathbf{y}}(y) / \sum_{y'} p_{C|y}(C|y')p_y(y')$.

The optimal loss of the classifier against a particular adversarial strategy is just a mixture of the optimal losses for each class list: $\sum_C p_{C|y}(C|y) \Pr[p_y(y)L^*(P|(y \in \{i, j\}), N, \mathcal{H})$.

If $p_{C|y}(C|y) = p_{C|y}(C|y')$ for all $y, y' \in C$, then $p_{y|C}(y|C) = p_{\mathbf{y}}(y) / \sum_{y' \in C} p_{\mathbf{y}}(y')$ and the adversary has not provided the classifier with extra information beyond the fact that $y \in C$. Thus $P_{x|y}P_{y|C} = P|(y \in C)$.

Now we can spcialize to the $m = 2$ case. Any stochastic matrix $s$ with zeros on the diagonal specifies an adversarial strategy for selecting $C$ with $p_{C|y}(\{i,j\}|i) = s_{i,j}$. Furthermore, if $s$ is also symmetric, $p_{C|y}(\{i,j\}|i) = p_{C|y}(\{i,j\}|j)$ and $p_{y|C}(i|\{i,j\}) = p_{y|C}(j|\{i,j\})$. Then the optimal classifier for the side-information game uses the $\binom{K}{2}$ optimal classifiers for the two-class games and incurs loss $\sum_{i,j} Pr[Y = i]a_{i,j}s_{i,j}$ where $a_{i,j} = L^*(P|(y \in \{i,j\}), \mathcal{H}, N)$. Because the diagonal entries of $a$ are all zero, there is always a maximizing choice of $s$ with a zero diagonal. Thus it is not necessary to include that constraint on $s$ when specifying the optimization. $\square$

*Proof of Theorem 4.* If $w = 0$, then the lower bound is zero and holds trivially. Otherwise, $\frac{1}{\mathbf{1}^T w}w$ forms a probability distribution over the vertices. Let $X \in \mathcal{V}^{\mathbb{N}}$ be a sequence of i.i.d. random vertices with this distribution. From this sequence, we define a random independent set as follows. Include $v$ in the set if it appears in the sequence $X$ before any of its neighbors in $\mathcal{G}$. If $v$ and $v'$ are adjacent, at most one of them can be included, so this procedure does in fact construct an independent set. The probability that $X_i = v$ is $\frac{w_i}{\mathbf{1}^T w}$ and the probability that $X_i$ is $v$ or is adjacent to $v$ is $\frac{((A+I)w)_v}{\mathbf{1}^T w}$. The first time that the latter event occurs, $v$ is either included in the set or ruled out. If $w_i > 0$, the probability that $v$ is included in the set is $\frac{w_v}{((A+I)w)_v}$ and otherwise it is zero. Thus the quantity $P(S$ in Theorem 4 is the expected size of the random independent set and $\mathcal{G}$ must contain some independent set at least that large. $\square$

# B  Fractional packing and covering

In this section, we record some standard definition in fractional graph and hypergraph theory (24).

Let $\mathcal{G} = (\mathcal{V}, \mathcal{E})$ be hypergraph and let $B \in \mathbb{R}^{\mathcal{E} \times \mathcal{V}}$ be its edge-vertex incidence matrix. That is, $B_{e,v} = 1$ if $v \in e$ and $B_{e,v} = 0$ otherwise. Let $p \in \mathbb{R}^{\mathcal{V}}$ be a vector of nonnegative vertex weights. Let $\mathbf{0}$ be the vector of all zeros and let $\mathbf{1}$ be the vector of all ones (of whichever dimensions are required).

A fractional vertex packing in $\mathcal{G}$ is a vector $q \in \mathbb{R}^{\mathcal{V}}$ in the polytope

$$q \geq \mathbf{0} \qquad Bq \leq \mathbf{1}.$$

The minimum weight fractional vertex packing linear program is

$$\min p^T q \quad \text{s.t.} \quad q \geq \mathbf{0} \qquad Bq \leq \mathbf{1}.$$

A fractional hyperedge covering in $\mathcal{G}$ for weights $p$ is a vector $z \in \mathbb{R}^{\mathcal{V}}$ in the polytope

$$q \geq \mathbf{0} \qquad B^T z \geq p.$$

The minimum weight fractional hyperedge covering linear program is

$$\min_z \mathbf{1}^T z \quad \text{s.t.} \quad z \geq 0, \quad B^T z \geq p.$$

These linear programs form a dual pair.

The independent set polytope for $\mathcal{G}$ is the convex hull of the independent set indicator vectors.

Let $\mathcal{G}'$ be a graph on the vertex set $\mathcal{V}$. A clique in $\mathcal{G}'$ is a subset of vertices in which every pair are adjacent. Let $\mathcal{C}$ be the set of cliques of $\mathcal{G}'$ and let $C \in \mathbb{R}^{\mathcal{C} \times \mathcal{V}}$ be the clique vertex incidence matrix. (In fact this construction can be interpreted as another hypergraph on $\mathcal{V}$.) A fractional clique cover in $\mathcal{G}'$ for vertex weights $p$ is a vector $z \in \mathbb{R}^{\mathcal{C}}$ in the polytope

$$q \geq \mathbf{0} \qquad C^T z \geq p.$$

Somewhat confusingly, the dual concept is called a fractional independent set in $\mathcal{G}'$. This is a vector $q \in \mathbb{R}^{\mathcal{V}}$ in the polytope

$$q \geq \mathbf{0} \qquad Cq \leq \mathbf{1}.$$

## C  Hyperedge Finding

One challenge in computing lower bounds for $0 - 1$ loss in the multi-class setting is that we need to find hyperedges in the conflict hypergraph. In this section, we will consider an $\ell_2$ adversary: $N(x) = \{x' \in \mathcal{X} | \, ||x' - x||_2 \leq \epsilon\}$ and describe an algorithm for finding hyperedges within the conflict graph.

We first note that for an $n$-way hyperedge to exist between $n$ inputs $\{x_i\}_{i=1}^n$, $\{x_i\}_{i=1}^n$ must all lie on the interior of an $n - 1$-dimensional hypersphere of radius $\epsilon$.

Given input $x_1, ..., x_n$ where $x_i \in \mathbb{R}^d$, we first show that distance between any two points in the affine subspace spanned by the inputs can be represented by a distance matrix whose entries are the squared distance between inputs. This allows us to compute the circumradius using the distance information only, not requiring a coordinate system in high dimension. Then we find the circumradius using the properties that the center of the circumsphere is in the affine subspace spanned by the inputs and has equal distance to all inputs.

We construct matrix $X \in \mathbb{R}^{d \times n}$ whose $i^{th}$ column is input $x_i$. Let $D \in \mathbb{R}^{n \times n}$ be the matrix of squared distances between the inputs, i.e., $D_{i,j} = ||x_i - x_j||^2$.

We first notice that $D$ can be represented by $X$ and a vector in $\mathbb{R}^n$ whose $i^{th}$ entry is the squared norm of $x_i$. Let $\Delta \in \mathbb{R}^n$ be such vector such that $\Delta_i = ||x_i||^2 = (X^T X)_{i,i}$. Then given that $D_{i,j}$ is the squared distance between $x_i$ and $x_j$, we have

$$D_{i,j} = ||x_i||^2 + ||x_j||^2 - 2\langle x_i, x_j \rangle,$$

which implies that

$$D = \Delta \mathbf{1}^T + \mathbf{1}\Delta^T - 2X^T X.$$

Let $\alpha, \beta \in \mathbb{R}^n$ be vectors of affine weights: $\mathbf{1}^T \alpha = \mathbf{1}^T \beta = 1$. Then $X\alpha$ and $X\beta$ are two points in the affine subspace spanned by the columns of $X$. The distance between $X\alpha$ and $X\beta$ is $\frac{-(\alpha-\beta)^T D(\alpha-\beta)}{2}$, shown as below:

$$\frac{-(\alpha - \beta)^T D(\alpha - \beta)}{2} = \frac{-(\alpha - \beta)^T (\Delta\mathbf{1}^T + \mathbf{1}\Delta^T - 2X^T X)(\alpha - \beta)}{2}$$
$$= \frac{-(0 + 0 - 2(\alpha - \beta)^T X^T X(\alpha - \beta))}{2}$$
$$= ||X\alpha - X\beta||^2.$$

Now we compute the circumradius using the squared distance matrix $D$. The circumcenter is in the affine subspace spanned by the inputs so we let $X\alpha$ to be the circumcenter where $\mathbf{1}^T \alpha = 1$. Let $e^{(i)} \in \mathbb{R}^n$ be the $i^{th}$ standard basis vector. The distance between the circumcenter and $x_i$ is $||X\alpha - Xe^{(i)}||^2$. From previous computation, we know that $||X\alpha - Xe^{(i)}||^2 = \frac{-(\alpha - e^{(i)})^T D(\alpha - e^{(i)})}{2}$. Since the circumcenter has equal distance to all inputs, we have

$$(\alpha - e^{(1)})^T D(\alpha - e^{(1)}) = \ldots = (\alpha - e^{(n)})^T D(\alpha - e^{(n)}). \tag{7}$$

Note that the quadratic term in $\alpha$ is identical in each of these expressions. In addition, $e^{(i)T} De^{(i)} = 0$ for all $i$. So equation 7 simplifies to the linear system

$$e^{(i)T} D\alpha = c \implies D\alpha = c\mathbf{1}$$
$$\implies \alpha = cD^{-1}\mathbf{1}$$

for some constant $c$. Since $\mathbf{1}^T \alpha = 1$, we have

$$1 = \mathbf{1}^T \alpha = c\mathbf{1}^T D^{-1}\mathbf{1}$$
$$\implies \frac{1}{c} = \mathbf{1}^T D^{-1}\mathbf{1}$$

assuming that $D$ is invertible. The square of the circumradius, $r^2$, which is the squared distance between the circumcenter and $x_1$, is

$$
\begin{aligned}
&\|X\alpha - Xe^{(1)}\|^2 \\
=&\frac{-(\alpha - e^{(1)})^T D(\alpha - e^{(1)})}{2} \\
=&e^{(1)T}D\alpha - \frac{\alpha^T D\alpha}{2} \\
=&c - \frac{c^2 \mathbf{1}^T D^{-1}\mathbf{1}}{2} \\
=&\frac{c}{2} \\
=&\frac{1}{2\mathbf{1}^T D^{-1}\mathbf{1}}.
\end{aligned}
$$

Therefore, assuming matrix $D$ is invertible, the circumradius is $\frac{1}{\sqrt{2\mathbf{1}^T D^{-1}\mathbf{1}}}$.

The inverse of $D$ can be computed as $\frac{\det D}{\operatorname{adj} D}$. Since $\alpha = cD^{-1}\mathbf{1}$, we have $\alpha = c\frac{\det D}{\operatorname{adj} D}\mathbf{1}$. As $r^2 = \frac{c}{2}$, constant $c$ is non-negative. Therefore, $\alpha \propto \frac{\det D}{\operatorname{adj} D}\mathbf{1}$.

When all entries of $\alpha$ are non-negative, the circumcenter is a convex combination of the all inputs and the circumsphere is the minimum sphere in $\mathbb{R}^{n-1}$ that contains all inputs. Otherwise, the circumsphere of $\{x_i | \alpha_i > 0\}$ is the minimum sphere contains all inputs.

After finding the radius of the minimum sphere that contains all inputs, we compare the radius with the budget $\epsilon$. If the radius is no larger than $\epsilon$, then there is a hyperedge of degree $n$ among the inputs.

# D  Experimental Setup

In this section, we describe our experimental setup. Our code for computing bounds is also available at `https://github.com/inspire-group/multiclass_robust_lb`.

**Datasets:** We compute lower bounds for MNIST (17), CIFAR-10, and CIFAR-100 (16). Since we do not know the true distribution of these datasets, we compute lower bounds based on the empirical distribution of the training set for each dataset.

**Attacker:** We will consider an $\ell_2$ adversary: $N(x) = \{x' \in \mathcal{X} | \|x' - x\|_2 \leq \epsilon\}$. This has been used in most prior work (5; 22; 29).

**LP solver:** For solving the LP in Equation (4), we primarily use the Mosek LP solver (3). When the Mosek solver did not converge, we default to using CVXOpt's LP solver (2).

**Computing infrastructure.** In order to compute lower bounds, we perform computations across 10 2.4 GHz Intel Broadwell CPUs. For adversarial training, we train on a single A100 GPU.

**Training Details**  For MNIST, we use 40-step optimization to find adversarial examples during training and use step size $\frac{\epsilon}{30}$ and train all models for 20 epochs. For CIFAR-10 and CIFAR-100, we use 10 step optimization to find adversarial examples and step size $\frac{\epsilon}{7}$ and train models for 100 epochs. For MNIST TRADES training, we use $\beta = 1$ and for CIFAR-10 and CIFAR-100, we use $\beta = 6$. Additionally, for CIFAR-10 and CIFAR-100, we optimize the model using SGD with learning rate and learning rate scheduling from (14). For MNIST, we use learning rate 0.01.

**Architectures used:**  For CIFAR-10 and CIFAR-100, we report results from training a WRN-28-10 architecture. For MNIST, we train a small CNN architecture consisting of 2 convolutional layers, each followed by batch normalization, ReLU, and 2 by 2 max pooling. The first convolutional layer uses a 5 by 5 convolutional kernel and has 20 output channels. The second convolutional layer also uses a 5 by 5 kernel and has 50 output channels. After the set of 2 convolutional layers with batch normalization, ReLU, and pooling, the network has a fully connected layer with 500 output channels followed by a fully connected classifier (10 output channels). In §E.8, we consider the impact of architecture on closing the gap to the optimal loss.

# E    Additional Experimental Results

In this Section, we provide additional experimental results to complement those from the main body of the paper. We organize it as follows:

- §E.1: We analyze a toy problem on 2D Gaussian data to get a better understanding of the impact of hyperedges on the optimal loss computation.
- §E.2: We investigate why higher degree hyperedges dp not have a large impact on lower bounds at lower values of $\epsilon$.
- §E.3: We show the runtime explosion at higher values of $\epsilon$ that makes it challenging for us to report optimal loss values.
- §E.4: Classwise L2 distance statistics and heatmaps for pairwise losses used to compute class-only lower bounds in main paper.
- §E.5: We provide results with standard PGD-based adversarial training, and show it is outperformed by Trades.
- §E.6: We provide results on the CIFAR-100 dataset.
- §E.7: We show lower bounds for a different set of 3 classes than the one considered in the main body. Main takeaways remain the same.
- §E.8: We ablate across larger neural networks to check if increasing capacity reduces the gap to optimal.
- §E.9: We attempt dropping examples from the training set that even the optimal clasifier cannot classify correctly in order to improve convergence.
- §E.10: We compute lower bounds on the test set for MNIST 3-class classification

## E.1    Results for Gaussian data

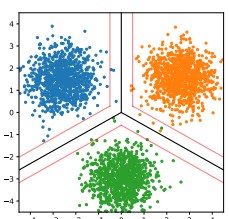

Figure 4: A sample 3-class Gaussian problem (each color pertains to a class) and a corresponding classifier for this problem shown in black. The classifier classifies a sample incorrectly when it lies over the edge of the $\epsilon$ margin (shown by the red lines) nearest the corresponding Gaussian center.

We begin with a 3-way classification problem on 2D Gaussian data. To generate our Gaussian dataset, we sample 1000 points per class from 3 spherical Gaussians with means at distance 3 away from from the origin (a sample is shown in Figure 4). We compute multiclass lower bounds via the LP in Theorem 1 on robust accuracy at various $\ell_2$ budget $\epsilon$ and display these values in Figure 5 as $L^*(3)$. Additionally, we compare to a deterministic 3 way classifier. This classifier is the best performing out of the 2 strategies: 1) constantly predict a single class (thus achieving $\frac{2}{3}$ loss) or 2) is the classifier in black in Figure 4 which classifies incorrectly when a sample lies over the edge of the nearest $\epsilon$ margin of the classifier.

We observe that at smaller values of $\epsilon$, the loss achieved by the 3-way classifier matches optimal loss ($L^*(3)$); however, after $\epsilon = 2.5$ for $\sigma^2 = 0.05$ and $\epsilon = 2.3$ for $\sigma^2 = 0.5$, we find the classifier no longer achieves optimal loss. This suggests that there is a more optimal classification strategy at these larger values of $\epsilon$. In Figures 6 and 7, we visualize the distribution of correct classification probabilities obtained through solving the LP with and without considering hyperedges. These figures are generated by taking a fresh sample of 1000 points from each class and coloring the point based on the correct classification probability $q_v$ assigned to its nearest neighbor that was used in the conflict

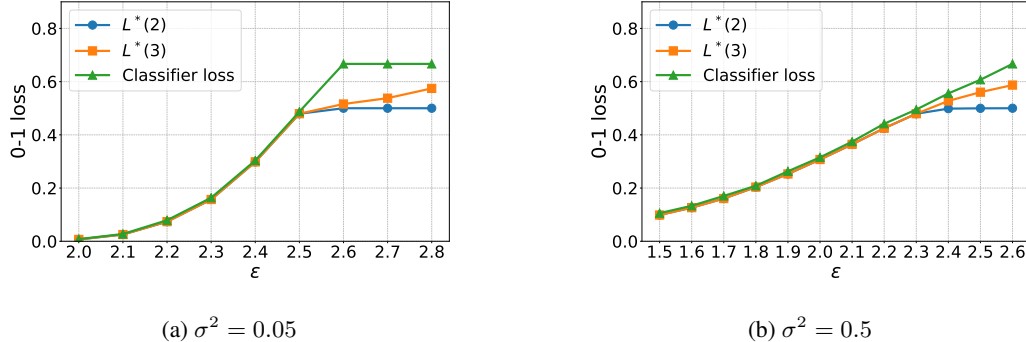

(a) $\sigma^2 = 0.05$           (b) $\sigma^2 = 0.5$

Figure 5: Lower bounds on error for the Gaussian 3-class problem ($\sigma^2 = 0.05$ and $\sigma^2 = 0.5$) computed using only constraints from edges ($L^*(2)$) and up to degree 3 hyperedges ($L^*(3)$) in comparison to the performance of the deterministic 3-way classifier depicted in Figure 4.

hypergraph when computing the lower bound. We observe from Figure 6, for all classes, the data are mostly assigned classification probabilities around 0.5. In Figure 7, we can see that when we consider hyperedges, we some of these 0.5 assignments are reassigned values close to $\frac{2}{3}$ and $\frac{1}{3}$. Interestingly, we notice that when we do not consider hyperedges, our solver finds an asymmetric solution to the problem (strategies for class 0, 1, and 2 differ) while when considering hyperedges this solution becomes symmetric.

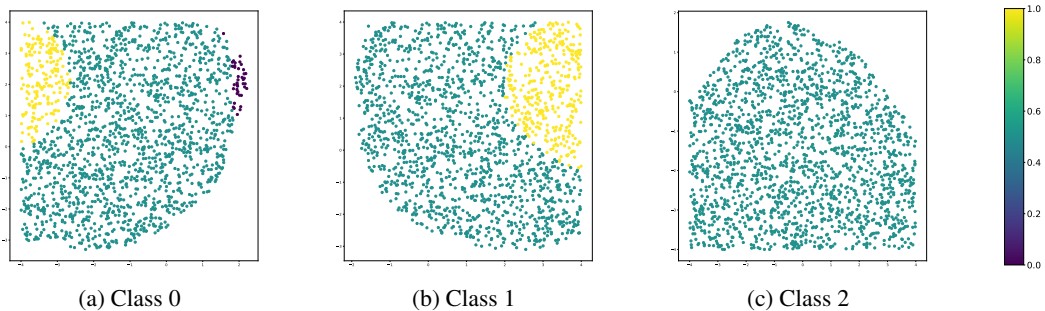

(a) Class 0        (b) Class 1        (c) Class 2

Figure 6: Distribution of optimal classification probabilities across samples from each class of the Gaussian obtained as a solution when computing $L^*(2)$.

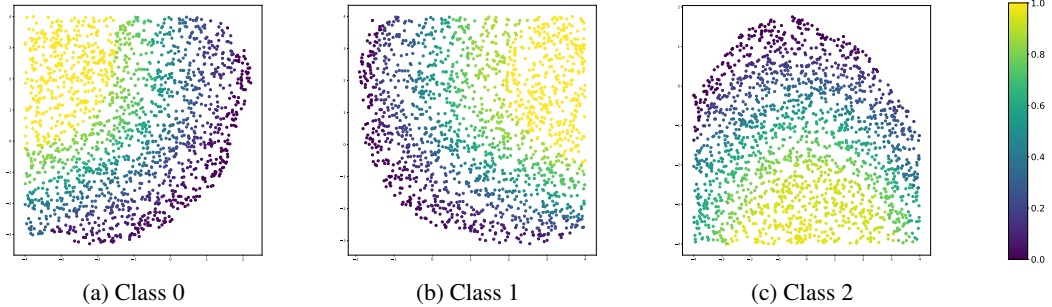

(a) Class 0        (b) Class 1        (c) Class 2

Figure 7: Distribution of optimal classification probabilities across samples from each class of the Gaussian obtained as a solution when computing $L^*(3)$.

## E.2 Impact of hyperedges

In Figure 8, we show the count of edges, degree 3 hyperedges, and degree 4 hyperedges found in the conflict hypergraphs of the MNIST, CIFAR-10, and CIFAR-100 train sets. We note that we did

not observe any increase in loss when considering degree 4 hyperedges at the $\epsilon$ with a data point for number of degree 4 hyperedges in Figure 8. We find that the relative number of edges and hyperedges is not reflective of whether we expect to see an increase in loss after considering hyperedges. For example in CIFAR-10, at $\epsilon = 4.0$, we there are about 100 times more hyperedges than edges, but we see no noticeable increase in the $0 - 1$ loss lower bound when incorporating these hyperedge constraints.

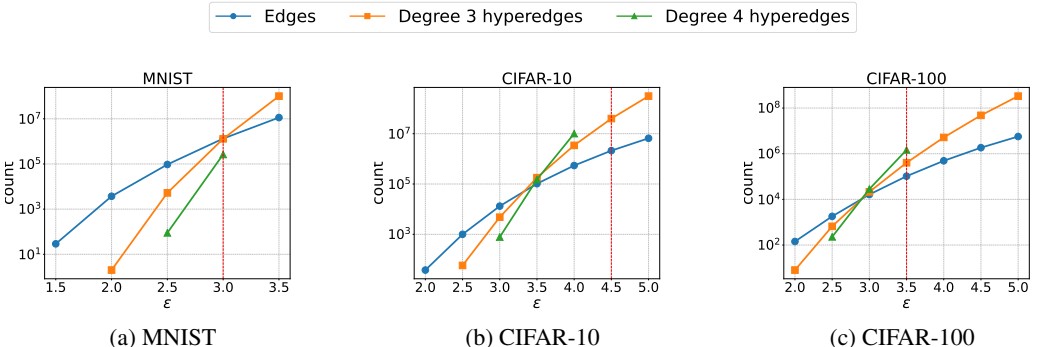

(a) MNIST           (b) CIFAR-10           (c) CIFAR-100

Figure 8: Number of edges, degree 3 hyperedges, and degree 4 hyperedges found in the conflict hypergraphs of MNIST, CIFAR-10, and CIFAR-100 train sets. The red vertical line indicates the $\epsilon$ at which we noticed an increase in the $0 - 1$ loss lower bound when considering degree 3 hyperedges.

To understand why including more information about hyperedges does not influence the computed lower bound much, we examine the distribution of $q_v$ obtained from solutions to the LP with ($L^*(3)$) and without degree 3 hyperedges ($L^*(2)$). Fig. 9 contains a histogram of the distributions of $q_v$ for MNIST. For small $\epsilon$, there is no change in the distribution of $q_v$, the distribution of $q_v$ is almost identical between $L^*(2)$ and $L^*(3)$. At larger values of $\epsilon$, in $L^*(2)$, a significant fraction of vertices are assigned $q_v$ near 0.5. While these shift with the addition of hyperedges, very few of them were in triangles of $\mathcal{G}^{\leq 2}$ that were replaced by a hyperedge in $\mathcal{G}^{\leq 3}$. This leads to the loss value changing minimally.

Similar to Figure 9, we plot the distribution of vertex weights $q_v$ obtained through solving the LP for $L^*(2)$ and $L^*(3)$ for CIFAR-10 in Figure 10. Similar to trends for MNIST, we find that the gap between $L^*(2)$ and $L^*(3)$ only occurs when the frequency of 0.5 weights is higher.

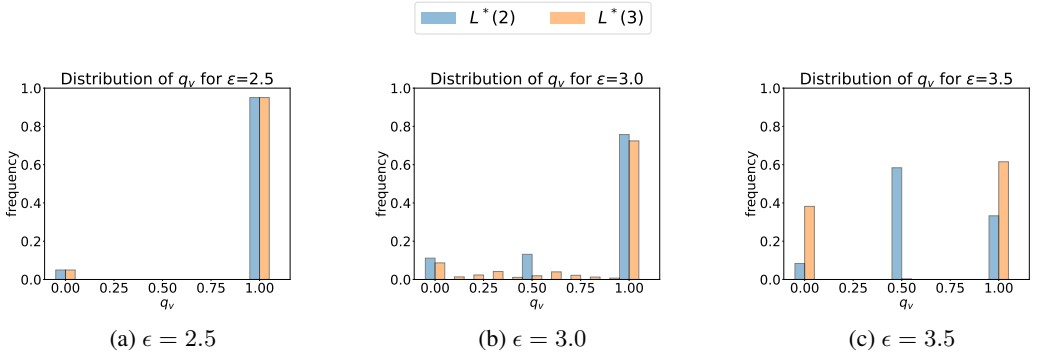

(a) $\epsilon = 2.5$           (b) $\epsilon = 3.0$           (c) $\epsilon = 3.5$

Figure 9: Distribution of optimal classification probabilities $q$ obtained by solving the LP with up to degree 2 hyperedges ($m = 2$) and up to degree 3 hyperedges ($m = 3$) on the MNIST training set.

### E.3 Computational complexity of computing lower bounds

Our experiments of $L^*(3)$ and $L^*(4)$ at higher $\epsilon$ are limited due to computation constraints. Figure 11 we see that the time taken to compute the $L^*(3)$ grows rapidly with $\epsilon$. We also report timings for all bounds for CIFAR-10 at $\epsilon = 4$ in Table 2. In future work, we are seeking algorithmic optimization to achieve more results at high $\epsilon$.

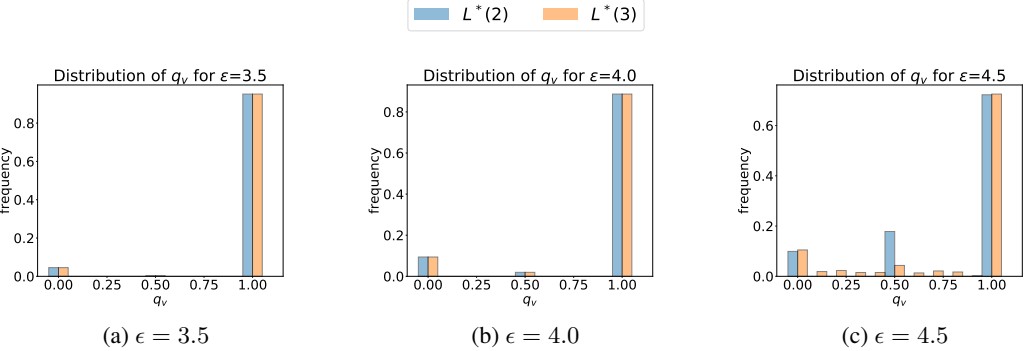

Figure 10: Distribution of optimal classification probabilities $q$ obtained by solving the LP with up to degree 2 hyperedges ($L^*(2)$) and up to degree 3 hyperedges ($L^*(3)$) on the CIFAR-10 training set.

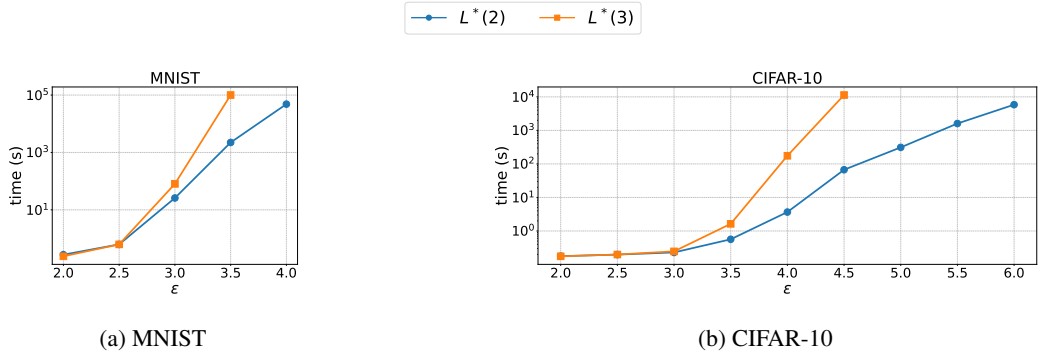

Figure 11: Time taken to compute $L^*(2)$ and $L^*(3)$ for MNIST and CIFAR-10.

### E.4 Classwise statistics and pairwise losses

In order to have a better understanding of the difficulty of the classification task under the presence of an $\ell_2$ bounded adversary, we report the average $\ell_2$ to the nearest neighbor of another class for each class in the MNIST and CIFAR-10 datasets in Table 3.

Another way of understanding the relative difficulty of classifying between classes is by computing the optimal loss for all 1v1 binary classification tasks. We note that these values are used in Section 4.2 to compute a lower bound on the optimal loss in the 10-class case from maximum weight coupling over optimal losses for $1v1$ binary classification problems. In Figure 12, we show the heat maps for optimal losses for each pair of 1v1 classification problems for $\epsilon = 3$ on MNIST and $\epsilon = 4$ on CIFAR-10. We find that for both datasets only a few pairs of classes have high losses. Specifically, for MNIST, we find that the class pairs 4-9 and 7-9 have significantly higher loss than all other pairs of classes. For CIFAR-10, we find that 2-4 has the highest loss compared to other pairs, and 2-6 and 6-4 are also quite high.

### E.5 Additional adversarial training results

In Figure 13, we also add the loss achieved by PGD adversarial training. We find that this approach generally performs worse on MNIST compared to TRADES and is also unable to fit to CIFAR-10 data at the $\epsilon$ values tested.

### E.6 Truncated hypergraph lower bounds for CIFAR-100

We provide results for truncated hypergraph lower bounds for the CIFAR-100 train set. We observe that similar to MNIST and CIFAR-10, including more hyperedge constraints does not influence the computed lower bound.

| Loss bound | Runtime (s) |
|---|---|
| $L^*(2)$ | 188.24 |
| $L^*(3)$ | 10413.91 |
| $L^*(4)$ | >86400 |
| $L^*_{co}(2)$ | 327.92 |
| $L_{CW}$ | 192.27 |

Table 2: Runtimes for computing different bounds for CIFAR-10 dataset at $\epsilon = 4$. We note that the $L^*_{co}(2)$ reports the time for computing all pairwise losses sequentially and can be sped up by running these computations in parallel. We also note that $L^*(4)$ computation did not terminate within a day.

| | 0 | 1 | 2 | 3 | 4 | 5 | 6 | 7 | 8 | 9 |
|---|---|---|---|---|---|---|---|---|---|---|
| MNIST | 7.07 | 4.33 | 6.94 | 6.29 | 5.72 | 6.29 | 6.42 | 5.52 | 6.29 | 5.15 |
| CIFAR-10 | 8.96 | 10.84 | 8.48 | 9.93 | 8.22 | 10.04 | 8.72 | 10.05 | 9.23 | 10.91 |

Table 3: Average $\ell_2$ distance to nearest neighbor in another class for each class in MNIST and CIFAR-10 datasets.

### E.7 Computed bounds for a different set of 3-classes

In Figure 15, we plot 3-class lower bounds via truncated hypergraphs ($L^*(2)$ and $L^*(3)$) for a different set of 3 classes as shown in the main body. These classes generally have less similarity than the classes shown in the main body of the paper causing the loss bound to increase more slowly as epsilon increases. However, we find that patterns observed for the 3 classes present in the main body of the paper are also present here: the gap between $L^*(2)$ and $L^*(3)$ is only visible at large values of 0-1 loss (ie. loss of 0.4).

### E.8 Impact of architecture size

Previously, we saw that adversarial training with larger values of $\epsilon$ generally fails to converge (leading to losses matching random guessing across 3 classes). We now investigate whether increasing model capacity can resolve this convergence issue. In Figure 16, we plot the training losses of 3 WRN architectures commonly used in adversarial ML research across attack strength $\epsilon$ for the 3-class CIFAR-10 (classes 0, 2, 8) problem. All models are trained with TRADES adversarial training. Interestingly, we find that the benefit of larger architecture size only appears for the smallest value of epsilon plotted ($\epsilon = 1$) at which the optimal classifier can theoretically obtain 0 loss. At larger values of epsilon, the improvement in using larger architecture generally disappears.

### E.9 Impact of dropping "hard" examples

From our experiments involving adversarial training, we found that training with large values of $\epsilon$ generally fails to converge. A potential way of trying to improve convergence using the results from our LP is to drop examples with optimal classification probability less than 1 and train with the remaining examples. Since even the optimal classifier cannot classify these examples correctly, these examples can be considered "hard" to learn. We find that in practice this does not lead to lower training loss; specifically, training without "hard" examples leads to a loss of 0.64 for CIFAR-10 with $\epsilon = 3$ while training with "hard" examples leads to a loss 0.57. We note that this loss is computed over the entire training dataset (including "hard" examples) for both settings.

### E.10 Lower bounds on test set

In the main text, we computer lower bounds on the train set as this would measure how well existing training algorithms are able to fit to the training data. In Table 4, we compute lower bounds on the test set (which contains 1000 samples per class) for MNIST 3-class classification between classes 1, 4, and 7. We find that the computed loss is similar to what is computed on a subset of the train set which contains the same number of samples per class.

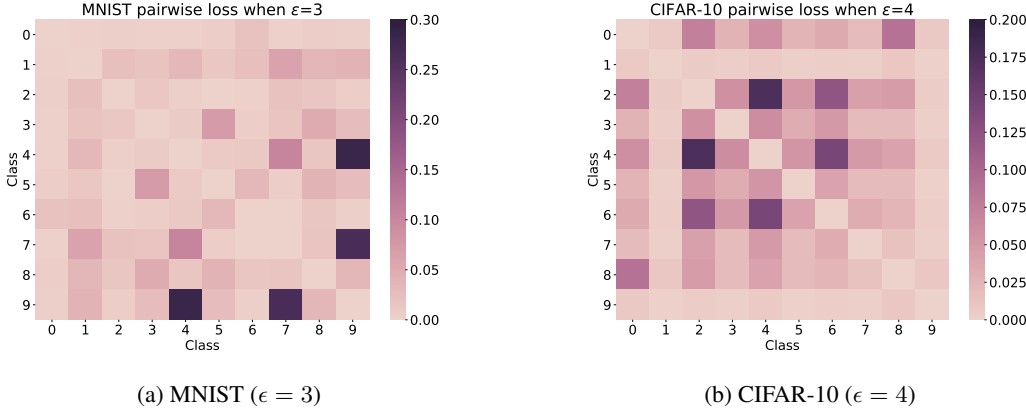

(a) MNIST ($\epsilon = 3$)

(b) CIFAR-10 ($\epsilon = 4$)

Figure 12: Heat maps for optimal loss for each pair of 1v1 classification problems.

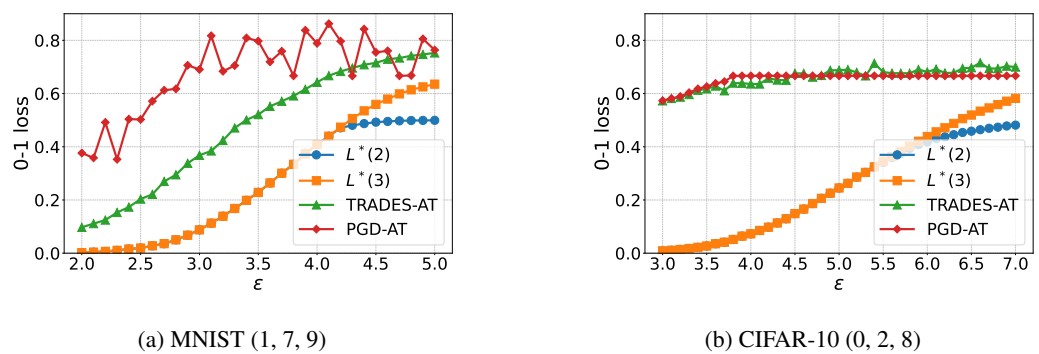

(a) MNIST (1, 7, 9)

(b) CIFAR-10 (0, 2, 8)

Figure 13: Lower bounds on error for MNIST and CIFAR-10 3-class problems (1000 samples per class) computed using only constraints from edges ($L^*(2)$) and up to degree 3 hyperedges ($L^*(3)$) in comparison to TRADES adversarial training (TRADES-AT) and PGD adversarial training (PGD-AT) loss.

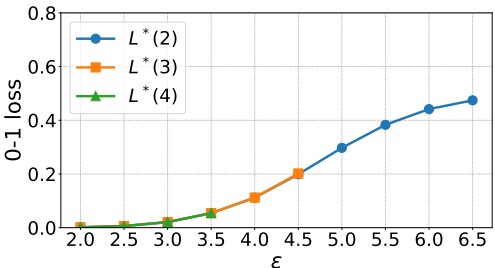

Figure 14: Lower bounds for optimal 0-1 loss the for CIFAR-100 train set

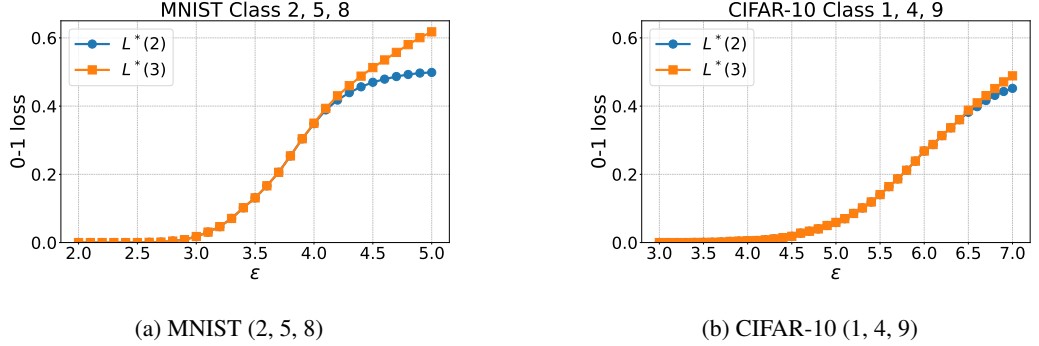

(a) MNIST (2, 5, 8)

(b) CIFAR-10 (1, 4, 9)

Figure 15: Lower bounds on error for MNIST and CIFAR-10 3-class problems (1000 samples per class) computed using only constraints from edges ($L^*(2)$) and up to degree 3 hyperedges ($L^*(3)$)

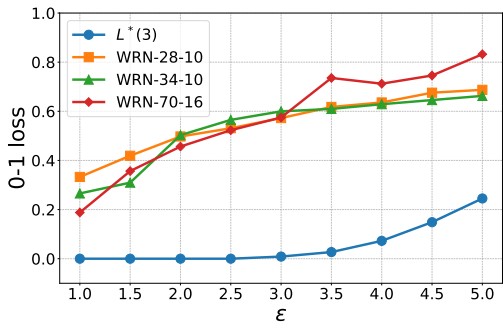

Figure 16: Impact of architecture size on training loss for 3-class CIFAR (0, 2, 8) at different strengths $\epsilon$ for models trained with TRADES adversarial training.

| $\epsilon$ | train set | train set (1000 samples per class) | test set |
|---|---|---|---|
| 2 | 0.0045 | 0.0020 | 0.0025 |
| 2.5 | 0.0260 | 0.0193 | 0.0149 |
| 3 | 0.1098 | 0.0877 | 0.0773 |
| 3.5 | 0.2587 | 0.2283 | 0.2181 |
| 4 | - | 0.4083 | 0.3987 |

Table 4: Optimal losses $L^*(3)$ for MNIST 3 class problem (classes 1, 4, 7) computed across the train set, the first 1000 samples of each class in the train set, and computed across the test set.

