# OpenReview forum: "Characterizing the Optimal $0-1$ Loss for Multi-class Classification with a Test-time Attacker"
_NeurIPS.cc/2023/Conference — NeurIPS 2023 spotlight_

### Official Review · Reviewer_Q9vC · 2023-06-20

**Soundness:** 4 excellent
**Presentation:** 4 excellent
**Contribution:** 3 good
**Rating:** 8
**Confidence:** 4

**Summary:**

The paper generalizes lower bounds on the adversarially robust error on a finite dataset from binary to multi-class classification.

**Strengths:**

1. The paper is well presented and easy to read, which is no mean feat for the amount of theory that is introduced and developed.
2. The formalization and assumptions are clearly stated, well arranged, and do not contain unnecessary complications.
3. The developed theory is correct as far as I can tell.
4. The experimental evaluations make sense for the discussed topic, regard standard baselines and are easy to understand. They are shown even while they don't support the necessity of the multi-class theory, which is nice to see.
5. The fact that calculating the bounds based on binary l2-ball intersections makes no difference in practice doesn't mean that the theory is not helpful for estimations that are in principle more correct.
6. Limitations are discussed honestly.
7. The code looks nice, but I haven't tested it.

**Weaknesses:**

1. While they are no clear weaknesses, there are some points unclear to me which I list below in the Questions section and I would much appreciate to see answered and in some cases discussed in the paper.
2. A substantial part of the formalization, theory and experimental design are not completely newly developed, but carried over from the previous paper "Lower Bounds on Cross-Entropy Loss in the Presence of Test-time Adversaries" which deals with the same problem in the more special case of binary classification. This is why I'm rating this submission currently as a standard "accept".

**Questions:**

1. While Equation (1) is true, I think it is not trivial, but a theorem that depends on some assumptions (which appear to be fulfilled by the considered hypothesis classes). It was proven in (Pydi,Jog 2022: "The Many Faces of Adversarial Risk")[https://arxiv.org/abs/2201.08956], but there might be earlier, more well known versions of the theorem. I think the intuition why (1) is true should be given.
2. l. 89 I think it should better read "the vector of _robustly_ correct classification"
3. l. 97 Should "feasible" read "achievable" here to stay consistent?
4. l. 97 Do the nonnegative linear combinations need to have factors that sum to less than 1?
5. Is L without * correct in Eq. (2)? If so, it wasn't defined.
6. l. 122 missing word
7. l. 124 extra (
8. Is the construction of the full hypergraph indeed computationally expensive for an l2 threat model? Intutitively, the geometry might make incidence matrix very sparse and its computation quite straightforward. Or is it rather an issue with the LP solver?
9. Would it make sense to regard the reverse truncation where if we find a triple of pairs with overlap (as in Fig. 1 left), we assume that there is also a point that generates the degree 3 hyperedge (and so on for higher degrees)? this might maybe yield an upper bound on $L^*(K)$.
10. l. 296 whether and in which sense the hypothesis class is much smaller is not obvious, and it is not clear if insufficient fitting within the class might be responsible for a large part of the gap.
11. In Figure 3, an empirical model evaluation as in Figure 2 should be included. ($L_{CW}$ is a bit confusing at first look, since many papers use that for Carlini-Wagner l2 attacks.)
12. Why not use the full AutoAttack?
13. The statistics on number of hyperedges should be included for all datasets, and a few values of epsilon (including 2 and 2.5), since they quantify the importance of regarding multi-class overlaps.
14. Also statistics like the average distance of an image to the closest one from another class would be nice to understand the geometry of the neighborhood overlaps.
15. It would be great if the authors could find a type of dataset and threat model where multi-class neighborhood overlap plays a bigger role than with MNIST and CIFAR. Maybe even a toy example would be illustrative.
16. Since the paper talks about optimal errors for finite distributions, it would make sense to show evaluations both on training and test sets.
17. A more concrete comparison to Trillos et al.[21], if applicable with evaluation numbers, would be helpful.
18. A discussion of the limitation to distributions with finite support and whether one can expect this assumption to be softened in future works building on this one would be interesting.

**Limitations:**

The limitations are discussed in detail.

---

> ### Author Rebuttal · Authors · 2023-08-09
>
> Thank you for your insightful feedback.  We are encouraged that you find our presentation clear and experiments interesting.  We address your questions below:
> 1) This is a good observation.  The technical issue with equation (1) is that for a particular $h$, the function $(x,y) \mapsto \sup_{\tilde{x} \in N(x)} \ell(h, (\tilde{x},y))$ may not be a measurable so the expectation may not be defined.  In this case, our hypothesis class does not help us, but our assumption that the data distribution has finite support ensures that the adversarial risk is well defined.
> One reason that we work only with finite-support probability distributions is to side-step these technical issues while still handling what we believe to be an interesting example of the problem.  Effectively we are placing the power-set sigma algebra on our space $\mathcal{X}$: every set and function becomes measurable at the cost of reducing the number of measures available.  This lets us avoid technical assumptions about the space $\mathcal{X}$ and the neighborhoods $N(x)$.
> Pydi and Jog provide some conditions to ensure that adversarial risk is well-defined, one of which is taking $\mathcal{X} = \mathbb{R}^d$ and working with Lebesgue measurable functions.  While we stated the theory portions of the paper in a more abstract setting, all of our experiments fit into that case.
> We should move our assumption that the data distribution is discrete from Section 2.2 up to 2.1 to justify equation (1).  We will also add a citation of Pydi Jog 2022 with a comment about the technical complexities that can arise in a more general setting.
>
> 2, 3, 5, 6, 7) We will update the paper to fix these typos.
>
> 4) When we optimize over the correct classification polytope, the weights are the example probabilities so they do sum to one. However, even if the weights did not sum to one, extending the region to include the origin would not affect the result of the optimization.
>
> 8, 13) Sparsity helps since we do not need to search all $\binom{n}{3}$ vertices for 3-way hyperedges, but even with sparsity, it is still expensive to find all hyperedges due to many triangles within the graph.  Solving the LP is also expensive at large $\epsilon$ due to the large number of constraints (we are generally bottlenecked by the LP solver inefficiency before being unable to compute hyperedges).  In Appendix Figure 5, we plot the number of edges, degree 3 hyperedges, and degree 4 hyperedges across $\epsilon$ for MNIST and CIFAR-10.  We find that the number of edges/hyperedges grows exponentially with the number of hyperedges increasing at a faster rate than edges.
>
> 9) This is a good suggestion and would lead to another upper bound on the optimal loss ($L^*(K)$).  The polytope that results from this process is the fractional independent set polytope of the conflict graph, which has a constraint for each clique.  Optimizing over this polytope gives the fractional independence number of the conflict graph.  In general, this polytope cannot always be computed quickly: there could be $\Omega(n^K)$ maximal cliques.
> However, we could probably compute the fractional independence number in many of the cases we experiment with, and are willing to add it to the updated version if the reviewer thinks it would be interesting.
>
> 10) Thank you for pointing this out, the gap may be due to optimization or due to the hypothesis class.  We will update the wording in this section to reflect this.
>
> 11) Thank you for the suggestion, we have updated Figure 3 to also include a line for PGD-AT performance as in Figure 2.  Please see our updated plot in the rebuttal pdf (Figure 1).  We have also shaded the region between the $L_{CW}$ and $L^*(2)$ lines to indicate the space where the true value of the optimal loss lies and make it more clear that $L_{CW}$ is an upper bound on optimal loss.
>
> 12) The second attack of the AA suite (APGD-T) uses targeted DLR loss which assumes at least 4 classes and cannot be used for 3 class experiments.  Additionally, we find that there is small change in robust accuracy when including all attacks so we choose to use APGD-CE in our experiments to reduce computation time.
>
> 14) Thank you for the suggestion.  We provide these statistics for each class in MNIST and CIFAR-10 in Table 1 of the rebuttal pdf.  We will add this into the Appendix.
>
> 15) Currently, it is unclear for what data distribution and threat models the multi-class overlap would play a bigger role.  In the Appendix, we present results for Gaussian data, but we observe similar trends as with MNIST and CIFAR-10 in this case.
>
> 16) Thank you for the suggestion, we provide some comparisons between evaluations on the training set and evaluations on the test set for MNIST classes 1, 4, and 7 in Table 3 of the rebuttal pdf.  We observe that the optimal loss computed on the test set is close to the optimal loss computed on a sample of the training set of the same size as the test set.  We will add this table to the Appendix of our paper.
>
> 17) The code used by Trillos et al. is unavailable and details about experimental setup are missing so it is difficult to compare.  Looking at the plot in Figure 6 of their paper, it seems they compute a loss lower bound of ~0.05 for MNIST classes 1, 4, 6, and 9 at $\ell_2$ budget of 800/255.  Computing $L^*(2)$ across the entire MNIST training set in this setting, we obtain $L^*(2)=0.21$.  We believe that Trillos et al. likely provide results on a subset of the dataset causing this discrepancy.
>
> 18) Thank you for this suggestion, we will add a discussion of this into the paper. We note that the optimal transport formulation proposed by Trillos et al. allows for general distributions, and focuses on transforming the problem to one where known methods exist to enable efficient bound computation. While our results are restricted to finite support, we are focused more on computing bounds on natural datasets efficiently by transforming the conflict graph directly.

---

> > ### Comment · Reviewer_Q9vC · 2023-08-21
> > **Response to Rebuttal**
> >
> > Thanks a lot for the detailed and insightful response!
> >
> > The additionally provided explanations and numbers are very helpful. I believe this is a strong contribution and am raising my score from 7 to 8.
> >
> > ---
> >
> > > 9.
> >
> > I personally would find upper bounds on the optimal lower bound very interesting, as they also enable judging the tightness of the lower bounds. I'd leave it to your judgement of the interestingness of these upper bounds and the actual numbers that can be calculated.
> >
> > > 16. (/15. evaluations both on training and test sets.)
> >
> > What I originally meant here was the question how close the AT model comes to the lower bounds on the training set. I think it would be interesting how much of a gap in training error could still be optimized away (if we consider the lower bounds to be close to the maximal lower bound) by better (or more overfitting) AT schemes, train attacks and models.

---

### Official Review · Reviewer_yCc4 · 2023-07-02

**Soundness:** 2 fair
**Presentation:** 1 poor
**Contribution:** 3 good
**Rating:** 6
**Confidence:** 2

**Summary:**

Deep learning techniques achieve state-of-the art performance on various classification tasks, but alarmingly, they are highly susceptible to adversarial perturbations. It is currently unknown whether there even exist classifiers that achieve low adversarial training risk on standard datasets. This paper aims to close this gap. The authors restate the adversarial learning problem over all classifiers as a linear program which can then be solved using standard LP techniques. This LP is stated in terms of the hyperedges of a hypergraph. As the resulting LP is computationally intractable, the authors then propose several ways to truncate this LP. The paper concludes with an experimental section in which they use the paper's techniques to upper bound the minimal possible adversarial loss for MNIST and CIFAR-10.

**Strengths:**

- It is currently unknown if finding robust classifiers to real-world datasets is possible. This paper presents a strong argument that such classifiers exist
- Using linear programming to attack this problem is very creative!

**Weaknesses:**

- I have some concerns about the correctness of this paper. Here are some specific issues
1. $q$ in line 20 of the supplementary material is not defined. This makes it quite hard to evaluate the correctness of the proof of Lemmas 1 and 2
2. I suspect Lemma 1 is false. Lower bounds on the adversarial risk computed in this paper rely on this lemma, so if Lemma 1 is false, these bounds would be invalidated.

Consider the following example: Consider 3 vertices with the incidence hypergraph of the left picture in Figure 1. Specifically, $\mathcal N(u)= \{u,v\},\mathcal N(v)=\{v,w\}, \mathcal N(w)=\{w,v\}$.  Then the incidence matrix $B$ is
|               |$u$|$v$|$w$|
|------------|-----|----|----|
|$e_{wu}$| 1   | 0   |1 |
|$e_{uv}$|1    | 1    |0|
|$e_{vw}$| 0    | 1   |1 |
|$e_{u}$  |1     | 0  |0 |
|$e_{v}$  |0     |0     |1 |
|$e_{w}$ |0     |1     |0|




Consider the vector $b= [0.5, 0.3, 0.2]$ (with vertices in order $u$, $v$, $w$). This vector clearly satisfies $b\geq 0$ and $Bb \leq \mathbf 1$.
We will now show that this vector is not in $\mathcal P_{\mathcal V, N,\mathcal H}$, as defined in line 96 of the paper.  For contradiction, assume that there is an $h$ for which $q_N(h)$ satisfies $0\leq b_i\leq q_N(h)_i$ for $i\in \{u,v,w\}$. By definition, $q_N(h)_v=\inf$ { $h(x): x\in N(v)$}.
Thus $h(u)\geq .5$ and $h(v)\geq .5$. As $h$ is a probability vector, it follows that $h=(.5, .5, 0)$ and $q_N(h)=(.5,0,0)$. This is a contradiction as $h(v)\leq b(v)=.3$.


3. In definition 2, h is defined to be $\cY$-valued but the expression $1-h(\tilde x,c)_y$ assumes that this function is $\mathbb R$ valued. As this definition is central to section 3, the correctness of this section is hard to evaluate
- The paper does not introduce central mathematical concepts or introduces them poorly. For instance,
1. line 82: "architecture" is discussed before neural nets are introduced
2. what is 'downwards-closed' in line 110?
3. The phrase "correct probability vectors" in Lemma 1 is misleading because the q's typically don't sum to 1
4. What does 'fractional coverings' mean in line 119?
5. lines 241-246: an argument is presented involving the fractional vertex packing polytope and the independent set polytope but these are only introduced very briefly

**Questions:**

- Can you explain the issues with Lemma 1 pointed in in the first bullet under weaknesses? Resolving this issue would convince me to change the review score
- Problems in optimal transport frequently need to solve linear programs in $\mathbf q$  with the constraint $\mathbf q\geq 0$. For computational expediency, the sinkhorn algorithm uses entropic regularization to deal with this constraint. Could you possibly make use of this technique?
- It seems that in the matrix inequality $Bq\leq 1$ in Lemma 1, many rows may be linearly dependent. Have you tried getting rid of such rows?

**Limitations:**

One central limitation of this work is that it studies the minimal possible adversarial risk over all possible (soft) classifiers rather than a particular function class.
This limitation is discussed in the paper.

---

> ### Author Rebuttal · Authors · 2023-08-09
>
> We thank the reviewer for their detailed engagement with our work and aim to address their concerns below, particularly those regarding the correctness of the main Lemmas in the paper.
>
> We are confident that Lemma 1 is correct. We show below that your suggested counterexample vector is achieved by an explicit classifier. We also walk through the constructions used in the proof for this example to provide some intuition for the general argument.
>
> 1. At line 20, $q$ is the vector that was introduced at line 15. It is an arbitrary point in $\mathcal{P}\_{m,V,N,\mathcal{H}_{soft}}$, the correct-classification-probability vector region. We will better explain the high level structure of the proof.
>
> 2. In your comment, you treated $h(u)$ and $h(v)$ as scalars, but they are in fact vectors in $\mathbb{R}^3$ and probability mass functions over $\mathcal{Y}$.
>   We have $q\_N(h)\_{(x,y)} = \inf \{ h(\tilde{x})\_y : \tilde{x} \in N(x) \}$ rather than $q\_N(h)\_{(x,y)} = \inf \{ h(x) : \tilde{x} \in N(x) \}$.
>   This confusion may be related to the typo discussed in point 3.
>
> For the left example in Figure 1, $\mathcal{P}\_{V,N,\mathcal{H}\_{soft}}$ contains the vector $b = (0.5,0.3,0.2)^T$: it is $q\_N(h)$ for the constant classifier $h(x) = b$.
> Applying the definitions, we have $q\_N(h)\_{(x,y)} = \inf \{ h(\tilde{x})\_y : \tilde{x} \in N(x) \} = \inf \{b\_y\} = b\_y$.
>
> We can work out the whole structure of $\mathcal{P}\_{V,N,\mathcal{H}\_{soft}}$ by following the arguments in the proof of Lemma 2.
> Lemma 1 states that contains the vectors $(q\_u,q\_v,q\_w)$ that satisfy the inequalities $q\_u \geq 0$, $q\_v \geq 0$, $q\_w \geq 0$, $q\_u + q\_v \leq 1$, $q\_u + q\_w \leq 1$, and $q\_v + q\_w \leq 1$.
> The latter three inequalities come from the three edges in the conflict graph.
> We will demonstrate the derivation of one of these as an example.
> Because the edge $\{u,v\}$ is present, there is some $\tilde{x} \in N(u) \cap N(v)$.
> Let $u$ be from class $0$ and $v$ be from class $1$.
> Thus we get inequalities $q\_N(h)\_u \leq h(\tilde{x})\_0$, $q\_N(h)\_v \leq h(\tilde{x})\_1$, and $h(\tilde{x})\_0 + h(\tilde{x})\_1 \leq 1$, which imply $q\_N(h)\_u + q\_N(h)\_w \leq 1$.
>
> The polytope described above has extreme points $(0,0,0)^T$, $(1,0,0)^T$, $(0,1,0)^T$, $(0,0,1)^T$, and $(1/2,1/2,1/2)^T$.
> The middle three points are the correct-classification-probability vectors of the three constant hard classifiers.
> The latter point is the correct-classification-probability vector of the classifier that assigns probability $\frac{1}{2}$ to each of whichever two classes could have produced $\tilde{x}$.
> This is described at lines 150-152 of the paper.
> This is not a constant classifier, which allows it to have better performance.
> We can achieve intermediate correct-classification-probability vectors (or better) by averaging the outputs of soft classifiers achieving the extreme points.
>
> We will attempt to make space to add the explicit descriptions of $\mathcal{P}\_{V,N,\mathcal{H}}$ to Section 2.3.
>
> 2. We have a typo at line 174. It should read $h: \mathcal{X} \times \binom{\mathcal{Y}}{m} \to [0,1]^{\mathcal{Y}}$, paralleling the definition at line 71. Thus $1 - h(\tilde{x}, c)\_y$ is a real number. We are sorry for the confusion that this caused.
>
> More:
>
> 1. We mean a function class in which each function is specified by a finite number of parameters. This encompasses neural networks but is more general. We will expand this discussion to improve the clarity.
> 2. The hyperedge set being downward closed means that if $e \in \mathcal{E}$ and $e' \subseteq e$, then $e' \in \mathcal{E}$. This follows from $\cap_{(x,y) \in e} N(x) \subseteq \cap{(x,y) \in e'} N(x)$, i.e. the same witness of the hyperedge e also witnesses e'.
> 3. The phrase should be read as correct-classification-probability vectors, because an entry of the vector is a correct classification probability. These are conditional probabilities (the entry $q_{N}(h)_v$ is the probability that the randomized classifier with distribution $h$ is correct given that the natural example is $v$), so as you point out, they do not sum to one in general. We can use this hyphenation.
> 4. A fractional covering is a standard concept from graph and hypergraph theory: it is a nonnegative weighting $z$ of the hyperedges such that each vertex receives coverage at least its weight. I.e. the sum of the weights of the hyperedges containing $v$ is at least the weight of $v$: $B^Tz \geq p$. We will add a textbook reference.
> 5. The discussion is brief due to space constraints. We will add a citation to make it clear that these are standard facts in combinatorial optimization and not new claims.
>
> Questions:
>
> * This point about entropic regularization connects to several other papers as well as directions for further research.
> If the inequality $q \geq 0$ is replaced with a cross-entropy regularization term $\sum_i p_i \log \frac{1}{q_q}$, which is infinite at the boundaries $q_i = 0$, the resulting problem is related to the optimal adversarial cross entropy loss.
> In Bhagoji et al. 2021, this is investigated for the two class setting.
> They compute the optimal cross-entropy loss by solving a sequence of 0-1 loss optimization problems.
> The dual problem of optimizing over adversarial strategies is more connected to optimal transport.
> Trillos et al. have multiple characterizations of the optimal loss in the multiclass setting terms of various optimal transport problems and use established entropy regularized solving methods to compute optimal losses.
> There are more possible ways to apply entropy regularization and we think that this is a fruitful direction for further investigation.
>
> * We do eliminate redundant constraints when possible, but it is not as simple as checking for linear dependence.
> Because $Bq \leq 1$ is an inequality, a row of $B$ can only be eliminated when it is a nonnegative linear combination of other rows and rows of $-I$ (which comes from the other constraint $-Iq \leq 0$).

---

> > ### Comment · Reviewer_yCc4 · 2023-08-10
> >
> > 1. I'm convinced by your explanation, I misunderstood the role of $h$. I am updating my score.
> > Consider including a proof outline of Lemma 1 in the main text of the paper, to further explain why this lemma would be true.
> >
> > Overall, I had a hard time with the exposition and organization of the technical portions of this paper. From the text, it is difficult to understand why the mathematical claims are true, (and sometimes also what they mean).
> >
> > 2. Your responses to these two questions helped me understand your approach. Consider including these discussions somewhere in your paper

---

> > > ### Author Response · Authors · 2023-08-11
> > >
> > > Thank you for your quick response.  We are happy that you are convinced about the correctness of Lemma 1.  We also appreciate your suggestions on adding a proof outline of Lemma 1 in the main text and adding discussions of the 2 questions to the paper.  Currently, due to space constraints we are unable to add a proof outline, but we plan on incorporating the discussion of the 2 questions into the Appendix of the camera-ready paper.
> > >
> > > On your comment about some mathematical claims being unclear, could you please elaborate on which claims you are referring to?  We are happy to clarify these points during the discussion period.

---

> > > > ### Comment · Reviewer_yCc4 · 2023-08-13
> > > >
> > > > Here are a handful of examples of text I had a hard time understanding.
> > > > - line 82: "architecture" is discussed before neural nets are introduced
> > > > -line 119: "fractional coverings" are not introduced
> > > > - lines 242-243: "fractional vertex polytope", "independent set polytope"
> > > > - line 251: "the conflict hypergraph is hereditary"
> > > > - line 261: "Cari-Wei bound"-- if this concept is important enough to title a section, I would expect an introduction and discussion of this result
> > > > -line 274: "In general, Lemma 4 can be thought of as a randomized procedure for rounding an arbitrary vector....". I'm not entirely sure what this means, or where rounding takes place in Lemma 4
> > > >
> > > >
> > > > A lot of this is due to not defining technical mathematical terms.
> > > > I would expect NeurIps submissions to clearly introduce mathematical concepts that are not well known in the ML community, even if they are standard within graph theory.

---

> > > > > ### Author Response · Authors · 2023-08-17
> > > > >
> > > > > Thank you for the feedback on improving paper clarity.  Due to space constraints, it is difficult to incorporate an introduction of all concepts into the main paper, but we will add additional discussion of graph theory concepts and Caro-Wei bound into the Appendix. We note that these graph theory concepts are not critical to the main contributions of the paper, but are present in discussions to give a graph theory interpretation.  We elaborate on the points you raised below:
> > > > >
> > > > > > line 82: "architecture" is discussed before neural nets are introduced
> > > > >
> > > > > By architecture, we mean a function class in which each function is specified by a finite number of parameters. This encompasses neural networks but is more general.  We will update the wording in this portion to reflect this idea more clearly.
> > > > >
> > > > > > line 119: "fractional coverings" are not introduced
> > > > >
> > > > > A fractional covering is a nonnegative weighting $z$ of the hyperedges such that each vertex receives coverage at least its vertex weight. I.e. the sum of the weights of the hyperedges containing $v$ is at least the weight of $v$: $B^Tz \geq p$.  We will add a textbook reference and further discussion in the Appendix.
> > > > >
> > > > > > lines 242-243: "fractional vertex polytope", "independent set polytope"
> > > > >
> > > > > We currently state the definitions of these polytopes briefly, although we are not explicit that both are subsets of $\mathbb{R}^{\mathcal{V}}$ just like $\mathcal{P}_{\mathcal{V},N,\mathcal{H}}$ is. We will write more explicit versions of these definitions and add a textbook reference.
> > > > >
> > > > > > line 251: "the conflict hypergraph is hereditary"
> > > > >
> > > > > By hereditary, we mean that the edge set of the conflict hypergraph is downward-closed (ie. if $e \in \mathcal{E}$ and $e' \subseteq e$, then $e' \in \mathcal{E}$).  We will update the text to add a definition of downward-closed in line 110 where it is first mentioned, and then change line 251 to state “edge set of the conflict hypergraph is downward-closed” instead of “conflict hypergraph is hereditary” for consistency.
> > > > >
> > > > > > line 261: "Cari-Wei bound"-- if this concept is important enough to title a section, I would expect an introduction and discussion of this result
> > > > >
> > > > > We will add further discussion of the Caro-Wei bound to this section. The original Caro-Wei lower bound on cardinality of the maximum independent set in a graph can be recovered by taking both w and p to be vectors of all ones. In this case, $((A+I)1)_v$ is one plus the degree of v.
> > > > >
> > > > >
> > > > > > line 274: "In general, Lemma 4 can be thought of as a randomized procedure for rounding an arbitrary vector....". I'm not entirely sure what this means, or where rounding takes place in Lemma 4
> > > > >
> > > > > Thank you for pointing this out, the sentence should say “the proof of Lemma 4 can be thought of as …”.  We will update the text to fix this.

---

### Official Review · Reviewer_QPxv · 2023-07-05

**Soundness:** 3 good
**Presentation:** 3 good
**Contribution:** 3 good
**Rating:** 7
**Confidence:** 3

**Summary:**

This work proposes to theoretically evaluate the robustness of a multi-class classifier by setting the lower and upper bounds of the optimal loss, i,e, the lowest loss achievable for a given hypothesis family. The lower bound is established by extending the conflict graph-based framework previously applied to the binary classification setting. The upper bound is built by generalizing the Caro-Wei bound. Beyond theoretical analysis, this work also focuses on computable methods to estimate the bounds, i.e. using the lower bounds of binary classification problems to compute that for the multi-class problem.




**Strengths:**

It is an important contribution to set up an upper and lower bound for the lowest achievable classification loss under the testing-time perturbation. These two bounds help to narrow down the possible range of the classification loss facing input noise, which measures accurately the robustness of a classifier.  Furthermore, this work provides the link between the optimal loss bound of binary classification tasks and that for multi-class tasks. This contribution enables efficient computation of the bound estimates when the number of classes increases.

**Weaknesses:**

I have to admit that I am not familiar with this theory. It takes me quite some time to read the contexts before I can figure out how this framework can be integrated into the investigated problem. I think for any readers / reviewers without the background information, offering even a brief introduction could be very helpful to evaluate the contribution.





**Questions:**

How tight are the upper and lower bound ?  Especially for the upper bound, this is established based the Caro-Wei bound, which is different from the conflict hypergraph framework. It is not clear how accurate the upper bound could be.

---

> ### Author Rebuttal · Authors · 2023-08-09
>
> We thank the reviewer for their positive appraisal of our paper and comments to improve it further. We address their comments below:
>
> **Further details to evaluate the contribution:**
> Our contribution lies in both theoretical and experimental aspects of characterizing the optimal robust 0-1 loss for multi-class classification. This optimal loss is important to characterize so that progress in defenses can be measured. There is value in knowing how well the best possible classifier could perform in a given adversarial setting so we know how far our current defenses are.
>
> In terms of a roadmap of our approach, we first provide an expression for the optimal loss and use Lemma 1 to connect the problem of finding the optimal loss with a linear program defined with respect to a graph of conflicts between data points from different classes (Section 2). Having established this connection, we then develop computationally more efficient methods to solve the conflict graph problem in practice (Section 3). We will elaborate upon our approach further in the beginnings of Sections 2 and 3.
>
> We extend the approaches taken in previous work [1,2,3] that characterize the optimal loss in the case of binary classification, to the multi-class setting. Our use of the conflict graph is inspired by [1,2] which first introduced this concept, and our major contribution lies in extending it to the notion of conflict hypergraphs for multi-class classification.
>
> **Tightness of the bounds:**
> We would like to clarify that the Caro-Wei upper bound uses the conflict graph to determine the size of the maximum independent set. However, the reviewer is correct to observe that this upper bound does differ from the form of the other bounds, which truncate the hypergraph to obtain a lower bound on the optimal loss. The tightness of the truncation based lower bounds as well as Caro-Wei upper bound depend directly on the structure of the conflict graph. In practice, we find that the bounds are tight when the perturbation budget epsilon is small. The gap between the bounds grows larger as we increase epsilon. An exact theoretical characterization of the gap is beyond the scope of this paper and we can add it to the limitations if the reviewer thinks that will add clarity.
>
> [1] A. N. Bhagoji, D. Cullina, and P. Mittal. Lower bounds on adversarial robustness from optimal transport. In Advances in Neural Information Processing Systems, pages 7496–7508, 2019.
> [2] A. N. Bhagoji, D. Cullina, V. Sehwag, and P. Mittal. Lower bounds on cross-entropy loss in the presence of test-time adversaries. In International Conference on Machine Learning, pp. 863-873. PMLR, 2021.
> [2] M. S. Pydi and V. Jog. Adversarial risk via optimal transport and optimal couplings. In Proceedings of the 37th International Conference on Machine Learning, pages 7814–7823, 2020.

---

### Official Review · Reviewer_baps · 2023-07-25

**Soundness:** 3 good
**Presentation:** 3 good
**Contribution:** 3 good
**Rating:** 6
**Confidence:** 3

**Summary:**

This paper aims to analyze the optimal 0/1 loss under the most strongest test time attack. The study commences by formulating the problem of obtaining the optimal classifier (based on 0/1 loss) as a linear program on a graph. Subsequently, the authors address the high computational complexity of calculating the optimal 0/1 loss by proposing a reduction technique through graph truncation. This reduction enables the computation of a lower bound of the 0/1 loss in a feasible timeframe. Ultimately, the authors present empirical evidence comparing their bound with the empirical defense method on real-world data. Notably, they discover that the widely-used baseline (adversarial training) still offers significant potential for improvement.

**Strengths:**

- The paper is written clearly and easy to follow.
- This paper extents the analysis on the optimal classifier under test time attack to multi-class classification setting. It seems to be a decent extension/contribution to the theoretical side of the field of adversarial robustness.
- Usually for this kind of problem, the computation complexity is one of the main challenge. However, they are able to find a way to speed it up. The idea of reducing it to a graph and then truncate the complex edges to reduce the computation needed for getting the bounds is interesting. In addition, they also show that empirically, such relaxation does not lose much information.

**Weaknesses:**

- Although it is mentioned in the related work that this is different from verifying robustness. I think it would be a valuable information to include bounds for verified classifiers in the empirical section. It would be interesting to see how close the existing verified classifiers are to the optimal bound.
- A related work titled "Robustness for non-parametric classification: A generic attack and defense." published in International Conference on Artificial Intelligence and Statistics, 2020 can be added. This work also utilize the idea of creating graph with the vertices being each example and edges being conflicting examples pairs. They tried to approach the optimal 0/1 by removing minimum number of edges. Although they did not compute specific bounds on the optimal 0/1 loss, I think it is still worth being discussed.
- This is a work with solid technical contribution. However, as mentioned in the limitation, the lack of implication on how to close the gap between the current robust classifiers and the optimal classifier limits the impact of this work to a moderate-to-high impact paper.
- Although a method for speeding up the algorithm through truncating the graph is proposed, the applicability of the proposed algorithm in practice seems to be still limited in practice due to heavy computational cost (in the experiment, only three class classification problems are run).


**Questions:**

Are there any of my review comments that misunderstood the paper? If so, please point them out. I am happy to adjust accordingly.


**Limitations:**

The authors do properly addressed the limitation of this work, which includes the lack of scalability of their algorithm and the lack of implications on how to close the gap between the current robust classifiers and the optimal classifier.

---

> ### Author Rebuttal · Authors · 2023-08-09
>
> We thank the reviewer for their insightful feedback and positive appraisal of our paper. We are glad they found it clear and easy to follow. We address their questions and concerns below:
>
> **Comparison to bounds for verified classifiers:** Thank you for the interesting prompt. We checked the available leaderboard (https://sokcertifiedrobustness.github.io/leaderboard/) on verifiably robust models for the settings we are concerned with ($\ell_2$ robustness for the MNIST and CIFAR-10 datasets), and found:
> - For MNIST, the best certifiably robust model has a 0-1 loss of 0.27 at a budget of 1.52 and 0.44 at a budget of 2.0.
> - For CIFAR-10, the best certifiably robust model has a 0-1 loss of 0.6 at a budget of 1.0 and 0.8 at a budget of 2.0.
>
> These are much higher than the optimal lower bound that is achievable for these datasets which is 0 in all these cases. We will add these numbers to the updated text in the paper for the value it provides in terms of providing a comparison to verifiably robust classifiers.
>
> **Related work on ‘Robustness for non-parametric classification’:** Many thanks for the pointer to this very interesting paper. Having gone through the paper, we find it to be quite relevant to our method for finding lower bounds. In particular, the construction of the graph in Section 3.1 matches that of our conflict graph. In addition, we have used a technique similar to the ‘adversarial pruning defense’ proposed in the paper to attempt to close the gap to optimal in Section D.9. of the Supplementary Material for neural networks, although we found little to no impact in the multi-class setting. Our technique was inspired by a similar one used in Bhagoji et al. (2021), which did find improvements in the two-class setting. We will update the related work to reflect the connection to this paper.
>
> **Potential measures to close the gap between optimal and current robust classifiers:** The main focus of our work is to provide a measure of progress for defenses by comparing them to the optimal loss. Regardless, we agree that it is interesting to consider measures to close the gap between optimal and current robust classifiers. In Section D.9. of the Appendix, we propose dropping hard data points to close this gap (inspired by Bhagoji et al. (2021)). However, we found limited to no improvement, pointing towards a need for a deeper exploration of the way in which the optimal loss construction can be used to close the gap. A few potential steps on the training side are increasing the architecture size and using additional unlabeled data. We could also potentially use the optimal classifier to overrule decisions in parts of the input space where trained neural networks are wrong, and the optimal classifier is fully specified. However, the input space coverage of the optimal classifier is low as it is only specified on points in the training data. Methods to improve this coverage would be an interesting direction for future work.
>
> **Results in the 10-class setting:** We would like to clarify that the paper does contain experiments beyond the 3-class setting. Section 4.2 of the paper has results and discussion for the 10-class case (See Figure 3). Due to computational limitations, we use a truncated version of the hypergraph containing up to degree-4 hyperedges. The Caro-Wei upper bound is also reasonably tight until $\epsilon=3.0$, indicating that the use of higher-order hyperedges will not provide any additional information about the optimal loss. We provide an updated version of Figure 3 in the attached pdf that may be clearer.

---

> > ### Comment · Reviewer_baps · 2023-08-12
> > **Thank you for responding to my concerns and questions.**
> >
> > After reading the responses, I still hold my original opinion that this is a technical solid paper and it can bring moderate-to-high impact on the development of theories for adversarial robustness. Therefore, I would like to maintain my original score.

---

### Author Rebuttal · Authors · 2023-08-09

We thank the reviewers for their thoughtful and constructive engagement with the paper. As reviewers ourselves, we greatly appreciate the reviewers’ efforts at providing thorough and insightful commentary on the paper. We have addressed all the reviewer’s concerns in the respective rebuttals, including providing Reviewer **yCc4** with a detailed explanation for why we strongly believe that Lemmas 1 and 2 are correct along with a refutation of the constructed counter-example.

We are glad that multiple reviewers found our theoretical contributions important (**baps**, **Qpxv**), approach creative (**yCc4**), and experiments interesting with easy to understand baselines (**Q9vC**).  We are also happy that multiple reviewers found the presentation to be clear and easy to follow (**baps**, **Q9vC**), appreciated our proposed techniques for making the problem more computationally efficient (**baps**, **Qpxv**), and appreciated our code and honest discussion of limitations (**Q9vC**).

In the attached pdf, we provide
1. An updated version of Figure 3 in the paper with losses from adversarial training (**Q9vC**) and shading to indicate the space between the upper bound ($L_{CW}$) and tightest lower bound where the optimal loss $L^*(10)$ would lie (**baps**, **Q9vC**)
2. A table of statistics for the average distance of examples to their nearest neighbor in another class for each class in MNIST and CIFAR-10 (**Q9vC**)
3. A table of optimal losses computed on the MNIST train set and MNIST test set (**Q9vC**)
We plan on incorporating both tables into the Appendix of our paper.

In light of suggestions for improvement suggested by the reviewers and in addition to the clarifications already provided in the rebuttals, we also commit to making the following changes to the camera-ready version of the paper:
1. Improvements to the clarity of the text:
    - Add an overview of our approach in Sections 2 and 3 (**QPxv**)
    - Add citations for graph theory concepts such as fractional coverings, fractional vertex packing polytope, and the independent set polytope (**yCc4**)
    - Fix typos pointed out by reviewers (**yCc4**, **Q9vC**)
2. Add discussion of the following papers:
    - Yang et al. 2020: “Robustness for non-parametric classification” (**baps**)
    - Pydi, Jog 2022: "The Many Faces of Adversarial Risk" (**Q9vC**)
3. A comparison to bounds for verified classifiers (**baps**)
4. Discussion of limitation to distributions with finite support (**Q9vC**)

---

### Decision · Program_Chairs · 2023-09-21

**Decision:**

Accept (spotlight)

**Comment:**

## Summary:

This theoretical paper studies the multi-class classification robustness. The authors prove upper and lower bounds of the lowest possible loss for a hypothesis class. The main results generalize lower bounds on the adversarially robust error on a finite dataset from binary to multi-class classification, which is an important problem in robustness. On the technical side, the authors use a link between robust error and certain graph theoretic concepts (e.g., the conflict hypergraph), which allows them to analysis the multi-class setting. Then, the authors can study the optimal loss using linear programming and a generalization of the Caro-Wei bound.

The reviewers found the contributions to be impactful, novel, and well-motivated. Along the way, the authors also study a computational approach to compute the optimal 0/1 loss. The reviewers also appreciated this result, as well as the empirical results that were possible because of it. In particular, it leads to new insights around adversarial training, which is one of the standard robust training algorithms in practice. The authors do experiments on 3- and 10-class problems to verify the theory in practice.

The authors and reviewers had a fruitful discussion. The authors did a great job quickly addressing many comments and incorporating them into the revision. For example, there were many comments raised around the exposition of the theoretical results, including increasing the self-contained nature of the paper by including definitions and adding more references.

Overall, after the author-reviewer discussions, all the reviewers speak highly of the paper and the impact of the results.

## Further improvements:

It would be good to continue improving the final version of the paper by including (either in the main paper or appendix) all of the necessary background. For example, there seem to be no theorems presented by the authors (only Lemmas and Corollaries). For a theoretical paper, it is usually good to present a main theorem or two, which are the key results for the readers to focus on.

The reviewers had a hard time full verifying the correctness of the paper, but they believe the results to be true and follow from the techniques in the paper. In particular, some concerns around the correctness of some lemmas were clarified during the author-reviewer discussion. On the other hand, the proofs in the appendix do seem somewhat terse, with many claims about graphs and probabilities that are stated with minimal justification. Given that there is no page limit for the appendix, it would be good to expand on the proofs of the lemma in Appendix A+B.